



# Discrete-wavelength DOAS NO$_2$ slant column retrievals from OMI and TROPOMI

Cristina Ruiz Villena[1], Jasdeep S. Anand[1], Roland J. Leigh[1,a], Paul S. Monks[2], Claire E. Parfitt[3], and Joshua D. Vande Hey[1]

[1]Department of Physics and Astronomy, Earth Observation Science Group, University of Leicester, Leicester, UK
[2]Department of Chemistry, University of Leicester, Leicester, UK
[3]Thales Alenia Space UK Ltd, Bristol, UK
[a]now at: EarthSense Systems Ltd, Leicester, UK

**Correspondence:** Cristina Ruiz Villena (crv2@leicester.ac.uk), Joshua Vande Hey (jvh7@leicester.ac.uk)

**Abstract.** The use of satellite NO$_2$ data for air quality studies is increasingly revealing the need for observations with higher spatial and temporal resolution. The study of the NO$_2$ diurnal cycle, global sub-urban scale observations, and identification of emission point sources are some examples of important applications not possible at the resolution provided by current instruments. One way to achieve increased spatial resolution is to reduce the spectral information needed for the retrieval, allowing both dimensions of conventional 2-D detectors to be used to record spatial information.

In this work we investigate the use of ten discrete wavelengths with the well-established Differential Optical Absorption Spectroscopy (DOAS) technique for NO$_2$ slant column density (SCD) retrievals. To test the concept we use a selection of individual OMI and TROPOMI Level 1B swaths from various regions around the world which contain a mixture of clean and heavily polluted areas. To discretise the data we simulate a set of Gaussian optical filters centred at various key wavelengths of the NO$_2$ absorption cross section. We perform SCD retrievals of the discrete data using a simple implementation of the DOAS algorithm and compare the results with the corresponding Level 2 SCD products, namely QA4ECV for OMI and the operational TROPOMI product.

For OMI the overall results from our discrete-wavelength retrieval are in very good agreement with the Level 2 data (mean difference < 5 %). For TROPOMI the agreement is good (mean difference < 11 %), with lower uncertainty owing to its higher signal-to-noise ratio. These discrepancies can be mostly explained by the differences in retrieval implementation. There are some larger differences around the centre of the swath and over water. While further research is needed to address specific retrieval issues, our results indicate that our method has potential. It would allow for simpler, more economic satellite instrument designs for NO$_2$ monitoring at high spatial and temporal resolution. Constellations of small satellites with such instruments on board would be a valuable complement to current and upcoming high-budget hyperspectral instruments.

## 1 Introduction

Nitrogen dioxide (NO$_2$) is a gaseous air pollutant from the NO$_x$ family (NO$_x$ = NO + NO$_2$) that exists in trace amounts in the atmosphere. Its sources are of natural origin (e.g. lightning, volcanoes, and microbial activity) or a result of anthropogenic



activities (e.g. agricultural biomass burning, fossil fuel combustion). While most of the background $NO_2$, produced mainly by natural processes, is in the stratosphere, in polluted areas tropospheric $NO_2$ is predominant. In these areas the main sources of $NO_2$ are anthropogenic emissions, which occur close to the surface in the boundary layer.

The most polluted regions are usually highly industrialised and densely populated urban areas, where the air pollution is complex due to the varied mix of constituents (Monks et al., 2009). There is evidence suggesting that $NO_2$ is a good proxy for the spatial variability of outdoor air pollution in urban environments (e.g. Levy et al., 2013), making it a suitable indicator of air quality.

$NO_2$ itself has harmful effects on human health, being associated for example with respiratory damage and premature death (WHO Regional Office for Europe, 2013). Moreover, it indirectly plays a role in the climate as it is a precursor of tropospheric ozone and aerosol, two of the Essential Climate Variables defined by the Global Climate Observing System (GCOS) (WMO, 2011).

There has been a continuous effort, particularly in recent decades, to regulate and monitor the concentrations of air pollutants such as $NO_2$ with the aim of: a) reducing emissions, and b) putting in place mitigation strategies to minimise the exposure of people to harmful levels. Nonetheless, despite a general decreasing trend on $NO_2$ concentrations in many locations across the globe, particularly in Europe and the United States (e.g. Castellanos and Boersma, 2012; Russell et al., 2012), the World Health Organisation (WHO) guidelines on air quality (WHO, 2006), and EU legislative limits in the case of Europe (EEA, 2018), are still often exceeded (e.g. DEFRA, 2018). In addition, in other areas of the world such as China and India concentrations of nitrogen oxides continued to rise until less than a decade ago (e.g. Huang et al., 2017; Richter et al., 2005; Hilboll et al., 2017). This highlights that there is still a lot to do to tackle the problem of air pollution and that a reliable, consistent long-term monitoring network is crucial.

There are two main methods for the continuous observation of $NO_2$ in the atmosphere: *in situ* measurements and space-borne remote sensing. *In situ* instruments such as chemiluminescence analysers (EPA, 1975; Dunlea et al., 2007) provide more accurate values because they directly measure the air they sample. However, it is not logistically or economically viable to install a large number of these around cities, so measurement points are usually sparse. On the other hand, satellite instruments provide global coverage but the spatial and temporal resolution is limited, e.g. 13 km × 24 km (at nadir) once per day for OMI (Levelt et al., 2006), and retrieving surface concentrations of $NO_2$ from satellite platforms is not straightforward. Increased spatiotemporal resolution is required to improve the accuracy of emission estimates and pollution forecasts (Ingmann et al., 2012).

$NO_2$ has typically been retrieved from measured Earthshine spectra using the well-established Differential Optical Absorption Spectroscopy (DOAS; Platt and Stutz, 2008) technique for over two decades, since the launch of GOME in 1995 aboard ERS-2 (Burrows et al., 1999). This was followed by SCIAMACHY (Bovensmann et al., 1999) aboard Envisat, OMI (Levelt et al., 2006) aboard Aura, GOME-2 (Munro et al., 2006) aboard MetOp-A, MetOp-B and MetOp-C, and more recently TROPOMI (Veefkind et al., 2012) aboard Sentinel-5P. Out of these, GOME-2, OMI and TROPOMI are still operational, and have a single daily overpass in the morning (GOME-2), or in the afternoon (OMI, TROPOMI). TROPOMI provides the best spatial resolution to date, with a nadir ground pixel size as small as 3.5 km × 5.5 km, or 1.8 km × 1.8 km in the occasionally





used zoom mode. Unlike their predecessors, OMI and TROPOMI have two-dimensional detectors that allow them to record multiple across-track viewing angles simultaneously (pushbroom measurement mode). While this mode results in higher spatial resolution, it comes at the cost of more optical complexity.

The DOAS principle relies on the separation of broadband and narrowband components of the reflectance spectrum and can resolve multiple gases simultaneously. DOAS retrievals typically use a few hundred spectral channels to perform the slant column density (SCD) fit for each ground pixel. The need for such a large number of channels requires complex optics, and careful wavelength calibration, for which usually the Fraunhofer lines in a reference solar spectrum are used. In addition, one dimension of the detector must be dedicated to recording all this spectral information.

One way to simplify instrument design and increase spatial resolution is to use a retrieval algorithm with reduced spectral information. The idea of using only a few discrete spectral channels to retrieve atmospheric trace gases has been used extensively for ozone retrievals. One example is the Total Ozone Mapping Spectrometer (TOMS; Heath et al., 1975), first launched in 1978 aboard Nimbus-7, which used pairs of discrete wavelengths in the Huggins band (310 - 340 nm) to retrieve ozone. Its strong, narrow absorption features and limited interference from other atmospheric gases makes ozone a relatively easy species to retrieve using discrete wavelengths. For a weak absorber like $NO_2$, it is more challenging, but it has also been done using passive techniques, such as the Brewer spectrometer (e.g. Cede et al., 2006; Wenig et al., 2008) and the Visible Nitrogen Dioxide instrument aboard the Solar Mesosphere Explorer (Mount et al., 1984), and active techniques, such as DIfferential Absorption LIDAR (DIAL; e.g. Hains et al., 2010). More recently, Dekemper et al. (2016) developed a new concept of "$NO_2$ camera" which employs pairs of wavelengths recorded sequentially using an acousto-optical tunable filter (AOTF) to image $NO_2$ in scenes containing plumes. However, these techniques rely on specific viewing geometries that make them unsuitable for nadir-viewing space applications.

In this work we explore the development, application and performance of a discrete-wavelength $NO_2$ retrieval algorithm based on DOAS (discrete-wavelength DOAS, DW-DOAS hereafter). Our approach combines the reduction in required spectral information with the advantages of DOAS in removing the effects of surface albedo, scattering, and interfering gases. We perform a feasibility study of the technique using data from OMI and TROPOMI and analyse the differences between our results and the operational Level 2 products. In addition, we discuss the implications of discretising DOAS and the potential application of our method to a future hypothetical instrument aimed at high-spatial-resolution urban air quality monitoring.

## 2 Method

### 2.1 Data sources

#### 2.1.1 Ozone Monitoring Instrument (OMI)

The Ozone Monitoring Instrument (Levelt et al., 2006) is an ultraviolet/visible (UV/VIS) spectrometer and operational since it was launched in 2004 aboard the NASA AURA spacecraft. It is a nadir-viewing instrument and follows a sun-synchronous polar orbit, with a daily local overpass time of 13.45 h. The visible band covers a spectral range of 350-500 nm, with a spectral



resolution of 0.63 nm and an average sampling distance of 0.21 nm. OMI has a nadir pixel size of 13 km × 24 km (13 km × 12 km in spatial zoom mode), with 60 across-track pixels covering a swath width of 2600 km.

OMI has had good radiometric performance so far, with only ~0.5 % decrease in the visible channel in the 15 years it has been operational (Levelt et al., 2018). However, it does have one main issue, known as the "row anomaly" (described in detail

in the KNMI OMI website: http://projects.knmi.nl/omi/research/product/rowanomaly-background.php), affecting the quality of the radiance in specific rows of the detector. OMI data from 2009 onwards is affected by this anomaly, although early signs started to be seen in 2007. In the work presented here we use Level 1B data from 2005, which is not affected by the row anomaly.

Several OMI Level 2 $NO_2$ products have been produced by different institutions (e.g. NASA (Krotkov et al., 2017), KNMI

(Boersma et al., 2011)). In this work we use the product released as part of the Quality Assurance for Essential Climate Variables (QA4ECV) project (Boersma et al., 2017), which includes recent improvements in the retrieval algorithm (Boersma et al., 2018; Zara et al., 2018).

### 2.1.2   TROPOspheric Monitoring Instrument (TROPOMI)

Launched in 2017 aboard ESA's Sentinel-5 Precursor, the TROPOspheric Monitoring Instrument (TROPOMI; Veefkind et al.,

2012) is the state of the art in remote sensing of atmospheric composition with heritage from OMI and SCIAMACHY. It is a pushbroom nadir spectrometer like OMI but it also covers the near infrared (NIR) and the shortwave infrared (SWIR). It flies in a sun-synchronous polar orbit with about the same daily local overpass as OMI. The visible band of interest in this study (band 4) covers a spectral range of 405-500 nm, with a spectral resolution of 0.55 nm and a spectral sampling of 0.2 nm. Along with an increased SNR, one of the major advantages of TROPOMI is its unprecedented spatial resolution of 3.5 km × 5.5

km, which goes down to 1.8 km × 1.8 km in zoom mode. Like OMI, the swath width is 2600 km, but TROPOMI has 450 across-track pixels. In this study we use TROPOMI Level 1B data and the operational $NO_2$ Level 2 product (van Geffen et al., 2018).

### 2.2   Data processing

### 2.2.1   Processing chain

We simulate discrete-wavelength data by discretising OMI and TROPOMI Level 1B data using digital Gaussian filters. In addition, the relevant absorption cross sections, solar reference and Ring spectrum, convolved with the corresponding row-dependent slit functions for either OMI or TROPOMI, are discretised. Before applying the filters, all the spectra are interpolated onto the radiance wavelength grid. This is done using the method employed by Bucsela et al. (2006) for the irradiance, and with a cubic spline interpolation for the other spectra. The spectral fit is performed using a custom-made DOAS retrieval routine

written in Python, using fitting parameters as close to those of the operational products as possible. The retrieval is described in more detail in section 2.3. Figure 1 shows a flow diagram of the processing chain.





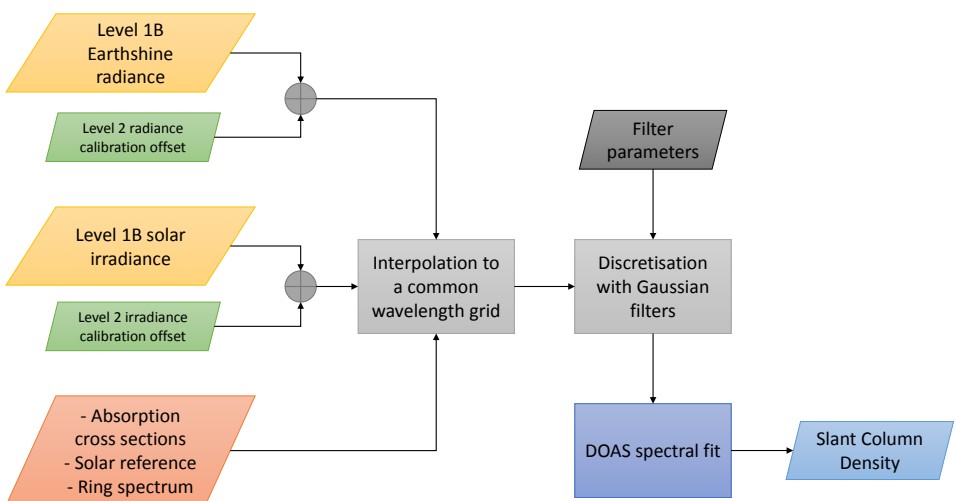

**Figure 1.** Flow diagram of the data processing used prior to the DW-DOAS retrieval.

### 2.2.2 Selection of the discrete channels

Earth radiance spectra are the result of a complex combination of absorption and scattering processes in the light path. This means that there is a great dependency on factors such as atmospheric composition, aerosol optical depth (AOD) and the reflective properties of the surface. When the available spectral information is limited to a few discrete points, the selection of
suitable channel parameters is critical for the performance of the retrieval.

In this work we have selected 10 channels in the 425-450 nm spectral region, centred at the wavelengths shown in Figure 2. This wavelength range has previously been used in SCIAMACHY (Bovensmann et al., 1999), GOME-2 (Munro et al., 2006), and some ground-based DOAS retrievals (e.g. Vandaele et al., 2005), because it contains strong $NO_2$ absorption lines. Each channel is modelled as a symmetric Gaussian function defined by three parameters: centre wavelength, full width at
half maximum (FWHM), and transmission peak. For this study, we consider only ideal filters (i.e. 100 % transmission), and a FWHM of 1 nm. The criteria for the wavelength selection applied in this work are as follows:

- *Select wavelengths at maxima and minima of the $NO_2$ absorption cross section, maximising the mean optical depth.*

- *Avoid wavelengths where there are large absorptions by interfering species.* While traditional DOAS solves this problem by fitting multiple species simultaneously, this benefit no longer exists when only a few discrete spectral points are
available. Water vapour, $O_2$-$O_2$ and the Ring spectrum represent the largest interferences in the spectral region of interest.

- *Minimise the total width of the spectral window.* DOAS retrievals can provide different results depending on the spectral window used in the fit (e.g. Alvarado et al., 2014). This owes to the fact that different spectral regions contain unique





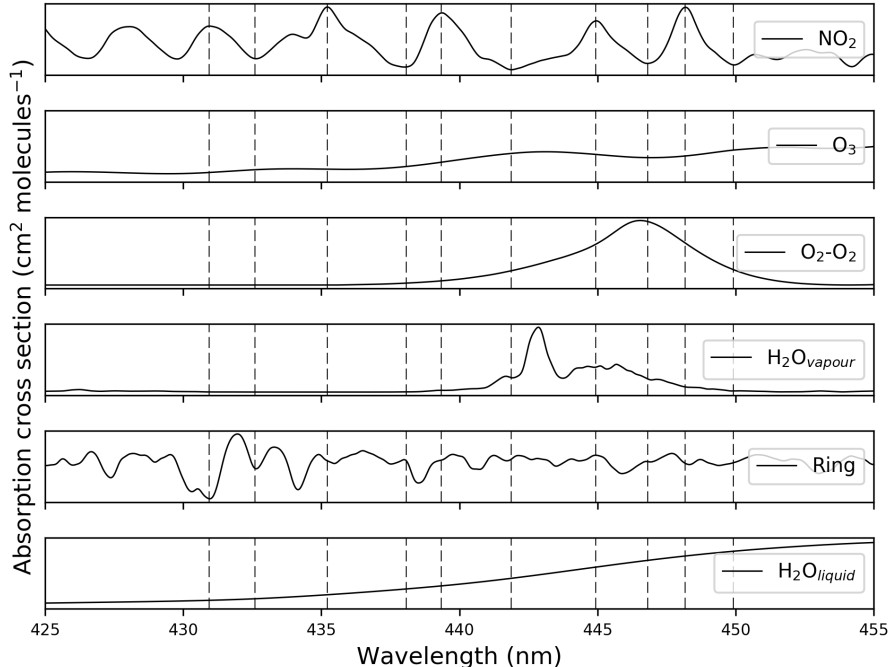

**Figure 2.** Absorption cross sections of relevant species (solid lines) and position of selected wavelengths (dashed lines) for this study.

features which might not be removed properly in the fit. For instance, Richter et al. (2011) found that when they increased the length of the fitting window to obtain higher SNR, unexplained spectral features appeared which were later shown to correspond to sand and liquid water signatures. This demonstrates that a short fitting window minimises the chances of unwanted spectral features. Moreover, it means that a lower order polynomial can be used in the DOAS fit.

## 2.3 Retrieval

### 2.3.1 Algorithm description

The retrieval algorithm used in this study is based on elements of DOAS (Platt and Stutz, 2008). There are different implementations of the DOAS technique, mainly the intensity fit (non-linear), and the optical density fit (linear). In DW-DOAS we use the linear approach to obtain the slant column density:

$$\sum_i \sigma_i(\lambda) \cdot N_{s,i} + P(\lambda) = -\ln\left(\frac{I(\lambda)}{I_0(\lambda)}\right) \tag{1}$$

where $\sigma_i$ is the absorption cross section of the $i$ species fitted, including the Ring spectrum as a pseudo-absorber (Chance and Spurr, 1997); $N_{s,i}$ are the slant column densities; $P(\lambda)$ is a low-order polynomial; $I(\lambda)$ is the Earth radiance; and $I_0(\lambda)$ is





the solar irradiance. Since this equation needs to be solved for each wavelength, the resulting problem is a system of equations. This can be represented with the following linear expression:

$$A \cdot x = B \tag{2}$$

$$
\begin{pmatrix}
\sigma_{NO_2}(\lambda_1) & \sigma_{O_3}(\lambda_1) & \cdots & 1 & \lambda_1 & \lambda_1^2 \\
\sigma_{NO_2}(\lambda_2) & \sigma_{O_3}(\lambda_2) & \cdots & 1 & \lambda_2 & \lambda_2^2 \\
\vdots & \vdots & \vdots & \vdots & \vdots & \vdots \\
\sigma_{NO_2}(\lambda_{10}) & \sigma_{O_3}(\lambda_{10}) & \cdots & 1 & \lambda_{10} & \lambda_{10}^2
\end{pmatrix}
\cdot
\begin{pmatrix} N_{s,NO_2} & N_{s,O_3} & \cdots & c & b & a \end{pmatrix}
=
\begin{pmatrix}
-\ln(\frac{I(\lambda_1)}{I_0(\lambda_1)}) \\
-\ln(\frac{I(\lambda_2)}{I_0(\lambda_2)}) \\
\vdots \\
-\ln(\frac{I(\lambda_{10})}{I_0(\lambda_{10})})
\end{pmatrix}
\tag{3}
$$

where $A$ is an $M \times N$ matrix containing the absorption cross sections and the polynomial basis for each wavelength, $x$ is a row vector of $N$ elements containing the slant column densities of the absorbers ($N_{s,i}$) and the polynomial coefficients ($a$, $b$, $c$), and $B$ is a column vector of $M$ elements containing the optical density for each wavelength. In order to solve for $x$ we calculate the pseudo inverse of matrix $A$, namely $A^{-1}$, using the Singular Value Decomposition (SVD) numerical method to factorise $A$:

$$A = U \cdot W \cdot V^T \tag{4}$$

$$A^{-1} = V \cdot W^{-1} \cdot U^T \tag{5}$$

where $U$ is an $M \times N$ column-orthogonal matrix, $W$ is an $M \times N$ diagonal matrix with non-negative real numbers in the diagonal (singular values), and $V$ is an $N \times N$ orthogonal matrix.

Although this approach is similar to a traditional DOAS retrieval, there are some differences arising from having only a few discrete spectral points. First, the order of the polynomial must not be greater than 2, as fitting higher order polynomials results in erroneously low slant column densities. In a way, this limits one of the key advantages of DOAS, this is, the ability to remove the broadband part of the reflectance. However, this can be overcome by having a fitting window narrow enough that the broadband component can be approximated by a 2nd-order polynomial, and is one of the criteria used in this work for wavelength selection (see Section 2.2.2).

Another consequence of discretising the spectra is that it is no longer possible to perform a wavelength calibration using the Fraunhofer lines of the solar reference spectrum. Therefore, no wavelength calibration is done as part of the retrieval. The implications of this limitation for a future operational instrument are discussed in Section 3.5.

The last difference between discrete and traditional DOAS is related to the ability to perform a "shift and squeeze" to correct for small spectral misalignments. In traditional DOAS two additional non-linear coefficients can be fitted to correct the spectra in this manner. This cannot be done in the context of a discrete-wavelength retrieval owing to the lack of spectral information available, so such parameters are not fitted.

Table 1 shows a comparison between our discrete-wavelength retrieval and the algorithms used for TROPOMI and OMI QA4ECV.





**Table 1.** NO$_2$ SCD retrieval details for OMI and TROPOMI reference products and DW-DOAS.

| | OMI QA4ECV (v 1.1) (Boersma et al., 2018) | TROPOMI (van Geffen et al., 2019a, b) | DW-DOAS (this work) |
|---|---|---|---|
| Fitting window | 405 - 465 nm | 405 - 465 nm | 430 - 450 nm |
| Fitting method | Optical depth (linear) | Intensity (non-linear) | Optical depth (linear) |
| $\chi^2$ minimisation method | Levenberg-Marquardt | Optimal Estimation | Not applicable |
| Level 1B uncertainty in $\chi^2$ | No | Yes | Not applicable |
| Selection reference spectrum | Annual mean (2005) solar reference | Daily solar reference | Same as reference product (annual mean/daily solar reference) |
| Polynomial degree | 4 | 5 | 2 |
| Intensity offset correction | Constant | No | No |
| Fitting parameters | O$_3$, NO$_2$, O$_2$-O$_2$, H$_2$O$_{vap}$, Ring, H$_2$O$_{liq}$, I$_{off}$, shift and stretch | O$_3$, NO$_2$, O$_2$-O$_2$, H$_2$O$_{vap}$, H$_2$O$_{liq}$, shift | O$_3$, NO$_2$, O$_2$-O$_2$, H$_2$O$_{vap}$, Ring, H$_2$O$_{liq}$ (only TROPOMI) |
| Treatment of Ring effect | Pseudo-absorber | Non-linear fit | Pseudo-absorber |
| Wavelength calibration (radiance) | Along with fit, 405-465 nm | Before fit, 405-465 nm | No |
| Temperature | 220 K | 221 K | Same as reference product (220 K/221 K) |

### 2.3.2 Retrieval uncertainty

The retrieval uncertainty is estimated using a method commonly employed as an independent evaluation of DOAS SCD uncertainty estimations (e.g. Zara et al., 2018; Boersma et al., 2007). The method calculates the uncertainty as the spatial variability of the SCD over a remote area in the Pacific Ocean which is considered to have background NO$_2$ concentrations. The assumption is that the variation in the NO$_2$ SCDs is caused solely by the retrieval uncertainty, therefore, this can be calculated as the standard deviation of spatial spread of SCDs. The area selected corresponds to latitudes between 60° S and 60° N, and longitudes between 150° W and 180° W. To account for light path differences the area is divided into 2° × 2° boxes so that the pixels in each box can be assumed to have similar path lengths. The geometric air mass factor (AMF) is used as an indicator of the path length for each pixel; a good description of AMFs can be found in Palmer et al. (2001). Boxes with high geometric air





mass factor (AMF) variability (> 5 %) are discarded. The relative AMF variability is calculated using the expression defined in (Zara et al., 2018):

$$AMF_{var} = \frac{\sqrt{\left(\overline{M_i^2} - \overline{M_i}^2\right)}}{M_i} \tag{6}$$

Where $M_i$ is the geometric AMF of each pixel ($i$) within one box, calculated as a function of the solar zenith angle ($\theta_s$) and the satellite viewing angle ($\theta_v$):

$$M_i = \sec\theta_{s,i} + \sec\theta_{v,i} \tag{7}$$

Then we calculate the deviation of each pixel from its box SCD mean and fit a Gaussian to the results, from which we obtain the standard deviation corresponding to the SCD uncertainty.

## 3   Results and discussion

### 3.1   OMI NO$_2$ SCD comparison

As an initial exercise, the NO$_2$ SCD results from our DW-DOAS retrieval of selected single orbits from January of 2005 are compared with the corresponding OMI QA4ECV NO$_2$ product. Figure 3 shows the results from both retrievals and the relative differences between them. The three orbits selected have a mixture of heavily polluted and clean areas with respect to NO$_2$. In all three swaths the datasets are highly correlated, with DW-DOAS generally producing lower NO$_2$ SCDs (~5 %). The largest differences are found around the centre of each swath, which coincides with the areas with the lowest SCDs in the QA4ECV dataset. These areas also are around the equator, where the geometric light paths are shortest. Furthermore, the middle of the swath is where ground pixel sizes are smallest, so the SCDs may be more susceptible to local variations in surface albedo.

It is visually apparent from Figure 3 that DW-DOAS results are slightly noisier than those retrieved by QA4ECV, particularly over unpolluted areas. This is expected given the limited spectral information available for the retrieval and indicates a lower sensitivity to NO$_2$ of DW-DOAS compared to hyperspectral retrievals. This difference in noisiness is quantified in the statistical uncertainty estimation (Section 3.3).

The differences between DW-DOAS and QA4ECV SCDs are normally distributed, as shown in the histograms in Figure 4. However, the negative biases indicate systematic differences between datasets, which are likely due to differences in retrieval implementation and settings. These biases are within the anticipated values from relevant sensitivity studies from the literature, which are summarised in Table 1. The main differences stem from the absence of wavelength calibration in the case of DW-DOAS, the inclusion of an intensity offset in the fit in the case of QA4ECV, and the differences in the fitting window. Also in Figure 4 are the correlation plots for all the selected swaths. These corroborate the good agreement between datasets that is evident in Figure 3, with $r > 0.99$ in all cases.





**Table 2.** Anticipated SCD differences between QA4ECV and DW-DOAS due to retrieval implementation differences, based on the literature.

|  | OMI QA4ECV | DW-DOAS | Anticipated SCD difference (QA4ECV - DW-DOAS) | Motivation |
|---|---|---|---|---|
| Fit window | 405-465 nm | 425-450 nm | $+0.5 \times 10^{15}$ molec. cm$^{-2}$ | Figure 11 in van Geffen et al. (2015) |
| Intensity offset | Yes | No | $\pm 0.3 \times 10^{15}$ molec. cm$^{-2}$ (depends on land vs ocean) | Figure 3(b) Boersma et al. (2018) |
| Wavelength calibration | Yes | No | $-0.85 \times 10^{15}$ molec. cm$^{-2}$ | van Geffen et al. (2015) |
| Net anticipated effect: | (QA4ECV - DW-DOAS) = $(0.5 + (\pm 0.3) - 0.85) \times 10^{15} = (-0.35 \pm 0.3) \times 10^{15}$ molecules cm$^{-2}$ | | | |

In order to check for any geographical and seasonal variabilities in the results we processed all single orbits from 4 days in January, April, July and October of 2005. The results can be seen in Figure 5, which shows the DW-DOAS retrieval results (scaled with the geometric AMF for clarity) and the relative differences with the QA4ECV product. Maps of the absolute differences can be found in Appendix A. Similar patterns to those seen in Figure 3 for individual orbits are also seen in the

global maps. The largest differences are seen mainly around the equator and they are highest in the April data. Some of the lowest differences are found in large plumes of NO$_2$, for example, in North America and China in the January map. Two other interesting features stand out from the global maps. Firstly, DW-DOAS seems to consistently underestimate the SCDs over the Sahara desert, which is likely due to the spectral signature of sand. It could be argued that it is the high albedo of the desert causing higher errors, but this only seems to happen significantly over that area. The second feature is found in South America

around 30° S, where there is an area of higher differences between retrievals. This is also apparent on the SCD maps, and seems to coincide with the region affected by the South Atlantic Anomaly (SAA), which is known to affect DOAS retrievals (e.g. Richter et al., 2011). The OMI QA4ECV product includes a spike correction that significantly reduces the scatter in the area affected by the SAA, which might explain the differences.

Figure 6 shows the correlation plots for DW-DOAS and QA4ECV using the global NO$_2$ SCD data from Figure 5. The agree-

ment between datasets when using global data is reduced owing to spatial features in the differences, as discussed previously. There are more outliers in the data from April and July, and most of them correspond to lower slant column densities and lower cloud radiance fractions.

### 3.2 TROPOMI NO$_2$ SCD comparison

To extend the analysis we performed a similar analysis to that done for OMI, described in section 3.1, using data from the

TROPOMI NO$_2$ operational product. First, the results from DW-DOAS for 3 selected orbits from 31/01/2019 are compared to the operational product (see Figure 7). As was the case for OMI, there is high correlation between the datasets, with DW-DOAS producing SCDs ~11 % smaller than TROPOMI. The largest differences are located towards the centre of the swath and

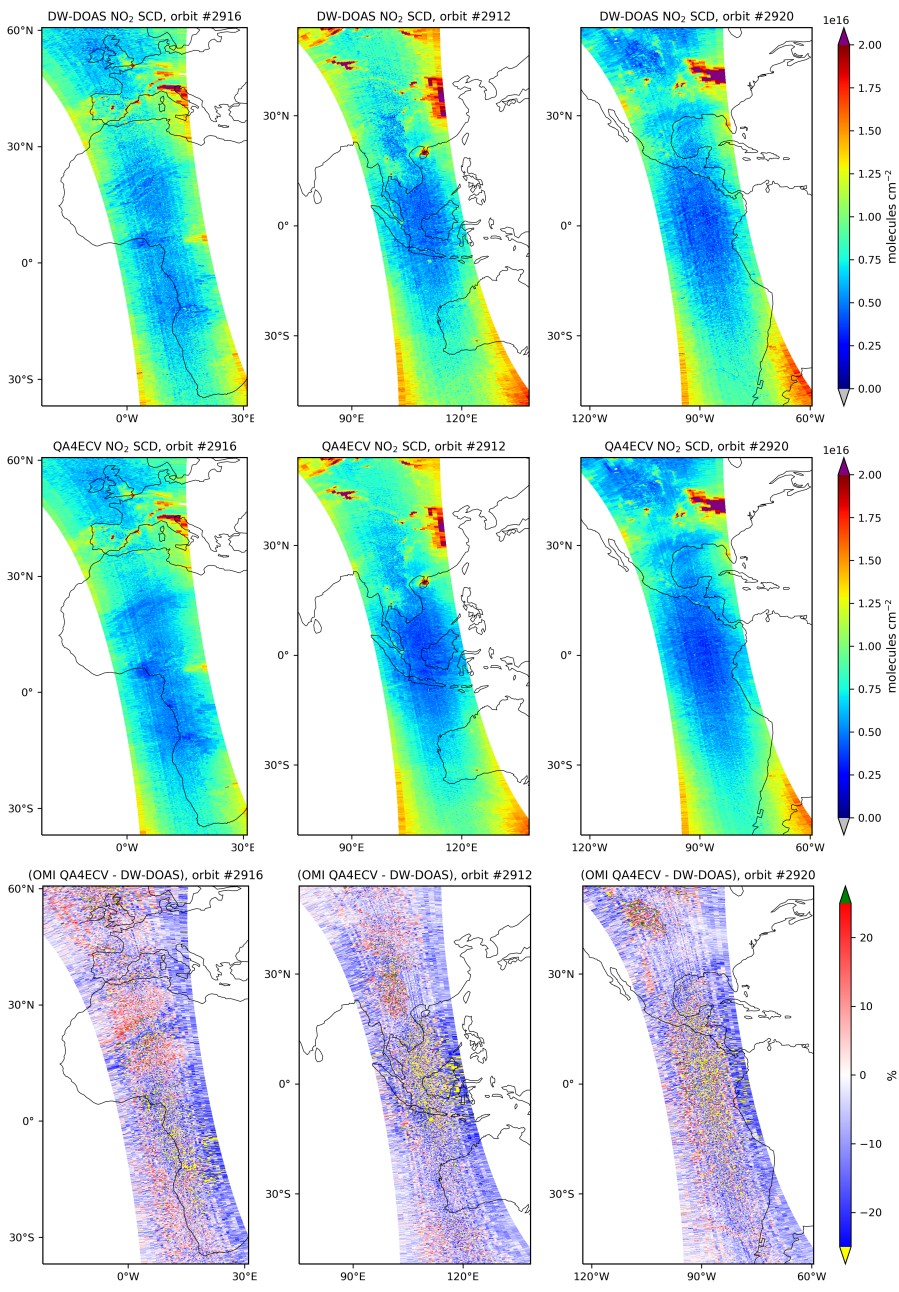

**Figure 3.** (top) NO$_2$ SCDs retrieved by DW-DOAS for selected orbits of 31/01/2005 in all-sky conditions, (middle) corresponding QA4ECV NO$_2$ SCDs, and (bottom) the relative differences between them calculated as (QA4ECV - DW-DOAS)/QA4ECV and expressed in %.

coincide with the lowest SCDs in the TROPOMI operational product. As with OMI data, the differences are smaller in areas with high SCDs.





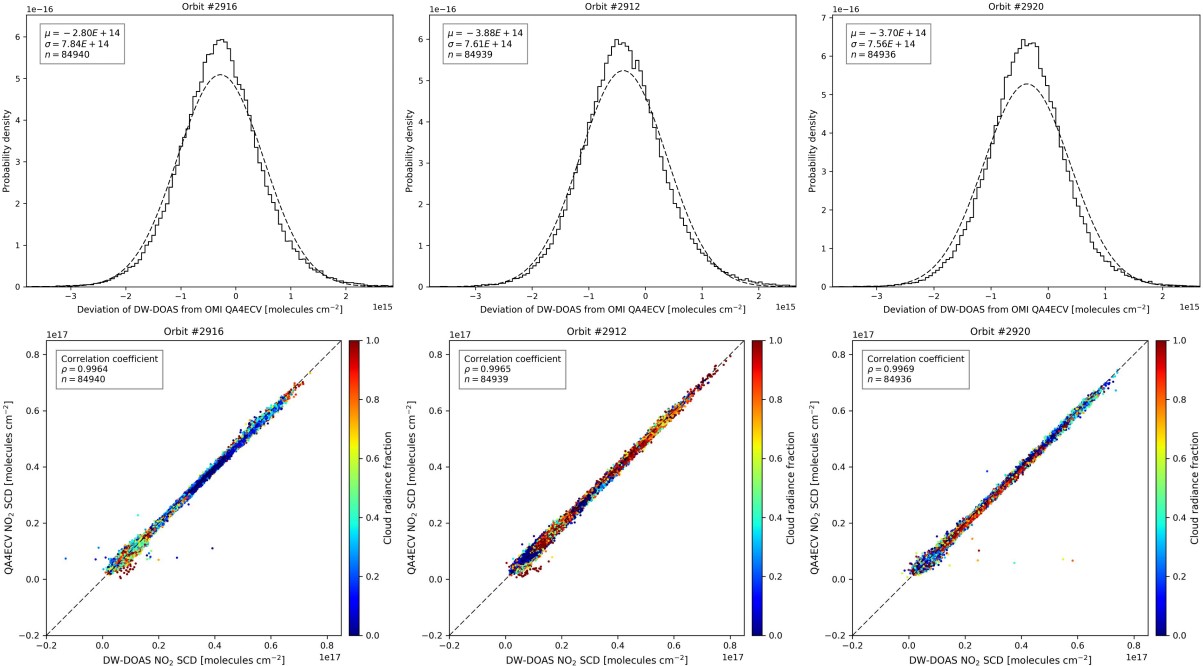

**Figure 4.** (top) Distribution of absolute differences between the NO$_2$ SCDs retrieved by DW-DOAS and QA4ECV for the orbits in Figure 3, calculated as (QA4ECV − DW-DOAS), and (bottom) corresponding correlation plot.

Figure 8 shows the histograms of the differences between retrieval results, and the correlation plots. These are similar to those obtained for OMI, and indicate that the differences are normally distributed and that the correlation is better than 0.99. However, in the case of TROPOMI the biases are much larger. Nonetheless, these still fall within the range of expected differences in SCD related to retrieval implementation and settings. Table 3 contains the expected range of SCD differences

according to the literature. The main contributions are from differences in fitting window, the inclusion of an intensity offset in the case of TROPOMI, and the different implementation of DOAS (non-linear in the case of TROPOMI, and linear in the case of DW-DOAS). Interestingly, although the biases are higher than in the case of OMI, the standard deviation of the differences between DW-DOAS and TROPOMI is smaller owing to its higher intrinsic SNR.

Figure 9 shows the global maps of SCDs retrieved by DW-DOAS and their relative differences with the TROPOMI opera-

10 tional product. Maps of the absolute differences can be found in Appendix A. The patterns in the single orbits are also seen throughout the global data. The largest differences are generally found in central across track pixels and are smaller at the edges of the swaths. While for OMI these were found around the equator, for TROPOMI they are spread further along the swaths, and are more pronounced over water. Interestingly, most of the areas with the largest differences coincide with high liquid water SCDs from the TROPOMI NO$_2$ level 2 operational product (retrieved as part of the DOAS fit; not shown here). It is unclear

what causes these spatial patterns, but surface albedo, smaller pixel sizes and viewing geometry might play a role. Over land

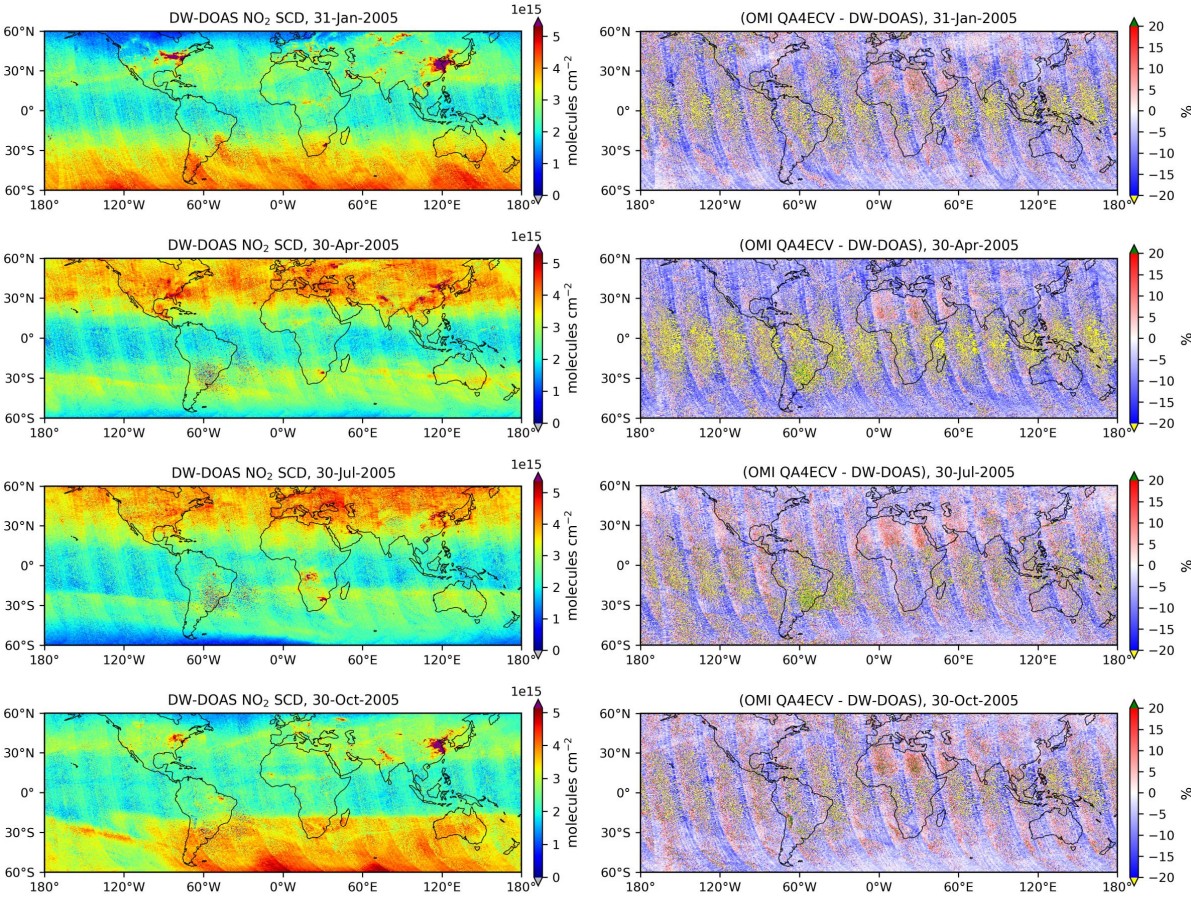

**Figure 5.** (left) Global DW-DOAS NO$_2$ SCDs scaled with geometric AMFs (for clearer data visualisation) for all single orbits of one day in January, April, July and October of 2005, and (right) the relative differences with QA4ECV calculated as (QA4ECV – DW-DOAS)/QA4ECV and expressed in %. The latitudes are limited to [60° S, 80° N]. Data from all-sky conditions have been used.

the differences are generally lower, with the exception of the Sahara desert. Unlike for OMI, for TROPOMI the effect of the SAA is not as obvious from the SCD maps, but it can be seen to a lesser extent in the maps of differences.

The global correlation plots in Figure 10 show similar correlation coefficients as those seen for OMI. However, once again the increased SNR is reflected in the standard deviation of the SCD differences, particularly for lower values, where it is 5 markedly smaller than that obtained for OMI. There are fewer outliers and, unlike for OMI, they mostly correspond to pixels with high cloud radiance fraction.

### 3.3 SCD uncertainty estimation

We apply the method described in Section 2.3.2 to calculate the NO$_2$ SCD statistical uncertainty for DW-DOAS for OMI and TROPOMI, including all the boxes in the region of interest for all four seasons. In order to validate our estimates, we also



**Figure 6.** Correlation plots of DW-DOAS and QA4ECV global NO$_2$ SCD data used in Figure 5.

apply the method to the reference datasets, namely the OMI QA4ECV and TROPOMI operational products, and compare the results. The calculations only include boxes with low geometric AMF variability (< 5 %). An example of the distribution of the deviation of the SCDs from their respective box mean for the January datasets is shown in Figure 11, and Table 4 contains the average results for all seasons. In all cases DW-DOAS gives higher uncertainty than the reference level 2 datasets, with this difference being more pronounced for OMI data. TROPOMI histograms have a better Gaussian fit, partly due to the higher quality of the data, but largely because its higher spatial resolution means there are more pixels for the same area used in the calculation, i.e. a larger sample size.



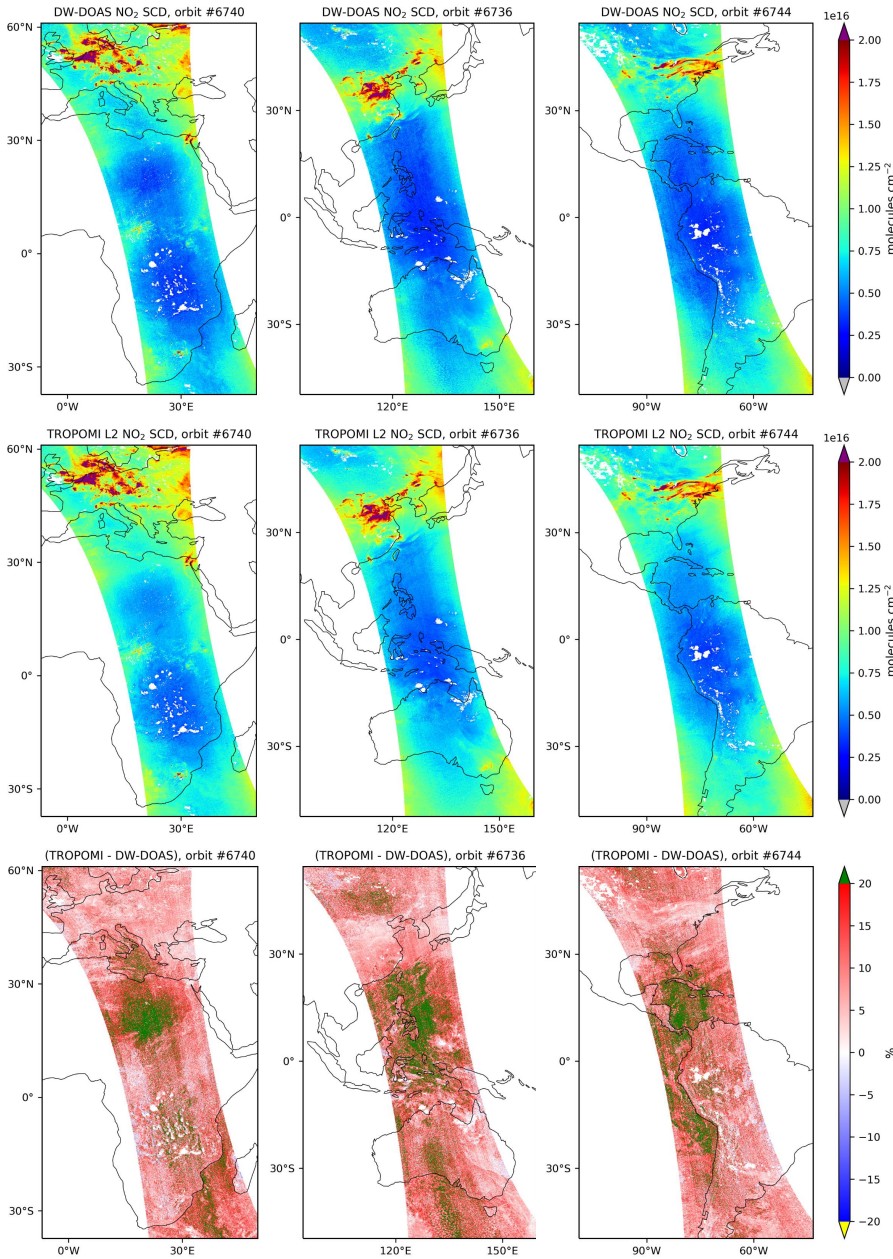

**Figure 7.** (top) NO$_2$ SCDs retrieved by DW-DOAS for selected orbits of 31/01/2019, (middle) corresponding TROPOMI NO$_2$ SCDs, and (bottom) the relative differences between them calculated as (TROPOMI - DW-DOAS)/TROPOMI and expressed in %. All the data are screened using the QA flag (qa > 0.5) from the TROPOMI NO$_2$ level 2 dataset, which includes all-sky pixels.

We also evaluate the sensitivity of DW-DOAS to striping, which is caused by the inhomogeneous illumination of the entrance slit of the instrument (Dobber et al., 2008). This issue is more pronounced in OMI, and it is usually corrected for after the DOAS





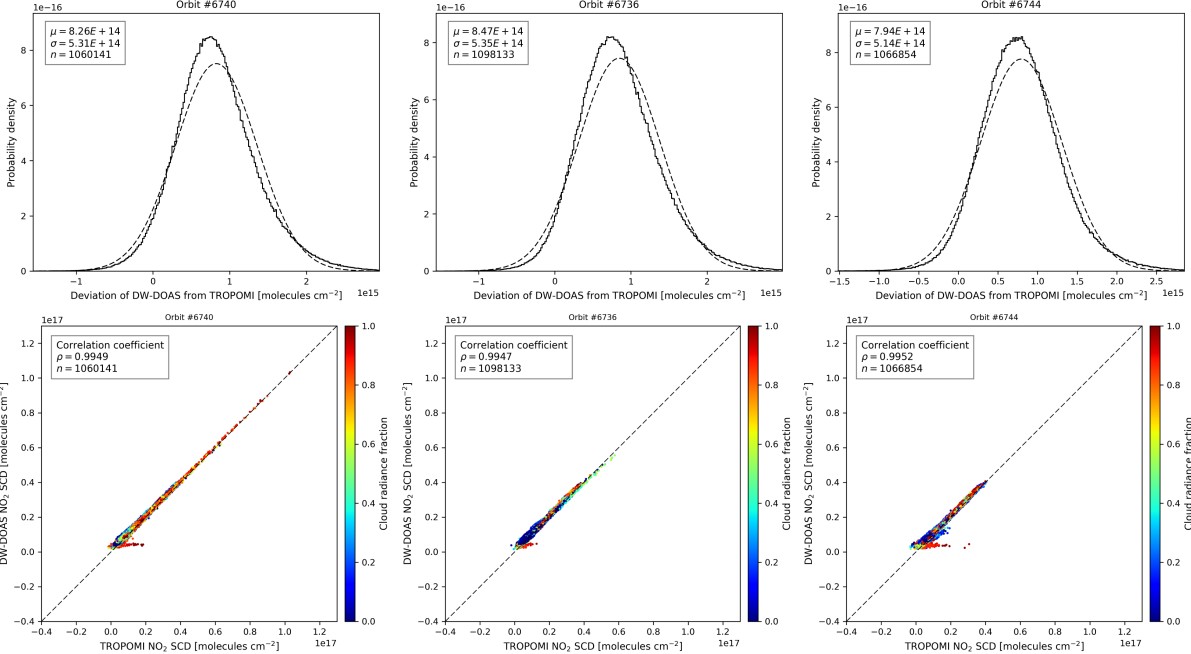

**Figure 8.** (top) Distribution of absolute differences between the NO₂ SCDs retrieved by DW-DOAS and TROPOMI for the orbits in Figure 7, calculated as (TROPOMI – DW-DOAS), and (bottom) corresponding correlation plot.

**Table 3.** Anticipated SCD differences between TROPOMI and DW-DOAS owing to retrieval implementation differences, based on the literature.

|  | TROPOMI | DW-DOAS | Anticipated SCD difference (TROPOMI - DW-DOAS) | Motivation |
|---|---|---|---|---|
| Fit window | 405-465 nm | 425-450 nm | $+0.5 \times 10^{15}$ molec. cm$^{-2}$ | Figure 11 in van Geffen et al. (2015) |
| Intensity offset | Yes | No | $-0.85 \times 10^{15}$ molec. cm$^{-2}$ | van Geffen et al. (2015) |
| Fit method | Intensity fitting | Optical density | $+0.2 \times 10^{15}$ molec. cm$^{-2}$ (over Africa) | Figure 3(a) Boersma et al. (2018) |
| Net anticipated effect: | (TROPOMI - DW-DOAS) = $(0.5 + (\pm 0.3) + 0.2) \times 10^{15} = (0.7 \pm 0.3) \times 10^{15}$ molecules cm$^{-2}$ | | | |

retrieval. Figure 12 shows the deviation from the mean SCD scaled with the geometric AMF as a function of across-track pixel number for one orbit over a clean area of the Pacific Ocean. The magnitudes of the peaks and troughs indicate that DW-DOAS has a higher sensitivity to striping compared to OMI QA4ECV, but it is less of an issue for TROPOMI because of the higher quality of the data.





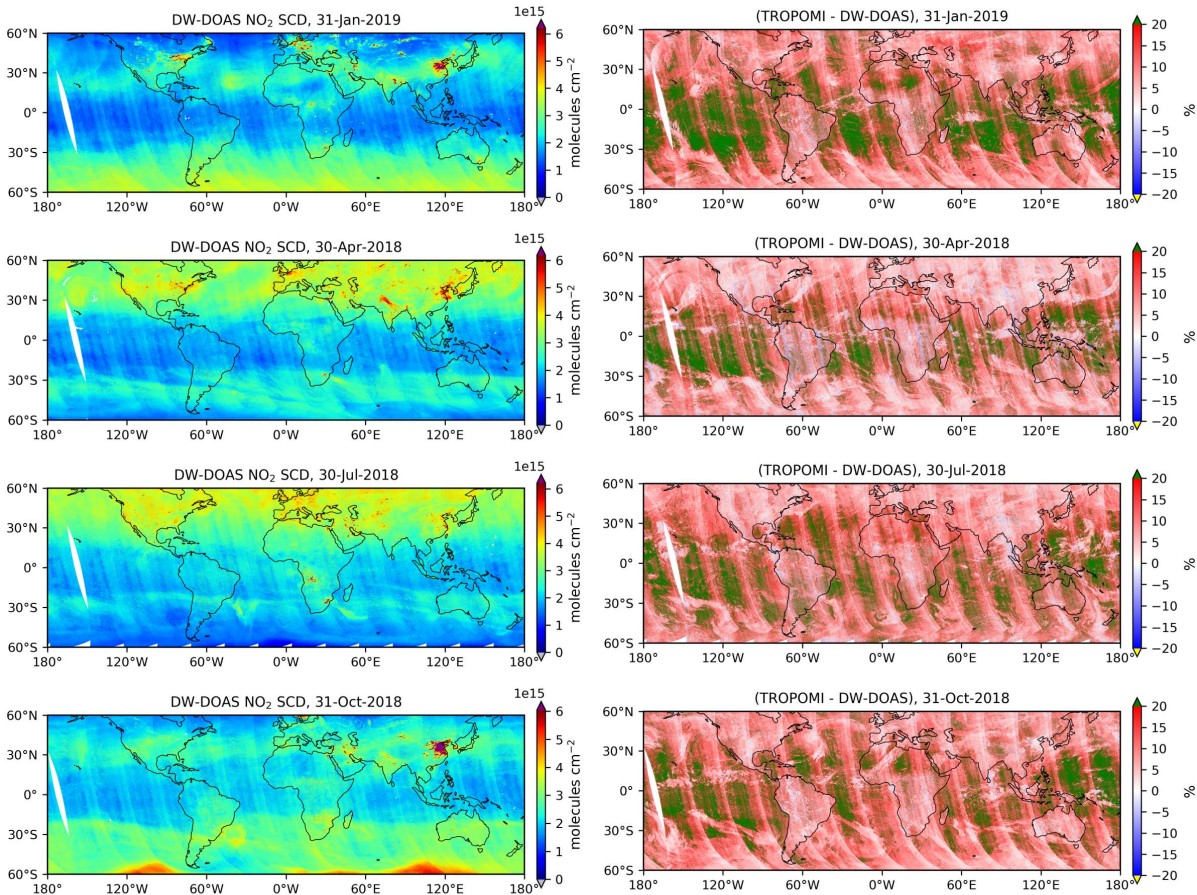

**Figure 9.** (left) Global DW-DOAS NO$_2$ SCDs scaled with geometric AMFs (for clearer data visualisation) for all single orbits of one day in January of 2019, and April, July and October of 2018, and (right) the relative differences with TROPOMI calculated as (TROPOMI – DW-DOAS)/TROPOMI and expressed in %. The latitudes are limited to [60° S, 80° N]. All the data are screened using the QA flag (qa > 0.5) from the TROPOMI NO$_2$ level 2 dataset, which includes all-sky pixels.

**Table 4.** Comparison of mean SCD statistical uncertainties for OMI QA4ECV, TROPOMI, and DW-DOAS, calculated from SCDs from a remote area in the Pacific Ocean within latitudes [60° S, 60° N] and longitudes [180° W, 150° W].

| Mean statistical $\sigma$ ($\times 10^{15}$ molec. cm$^{-2}$) | | |
|---|---|---|
| Instrument | DW-DOAS | Reference L2 product |
| OMI | 0.97 | 0.71 |
| TROPOMI | 0.68 | 0.54 |





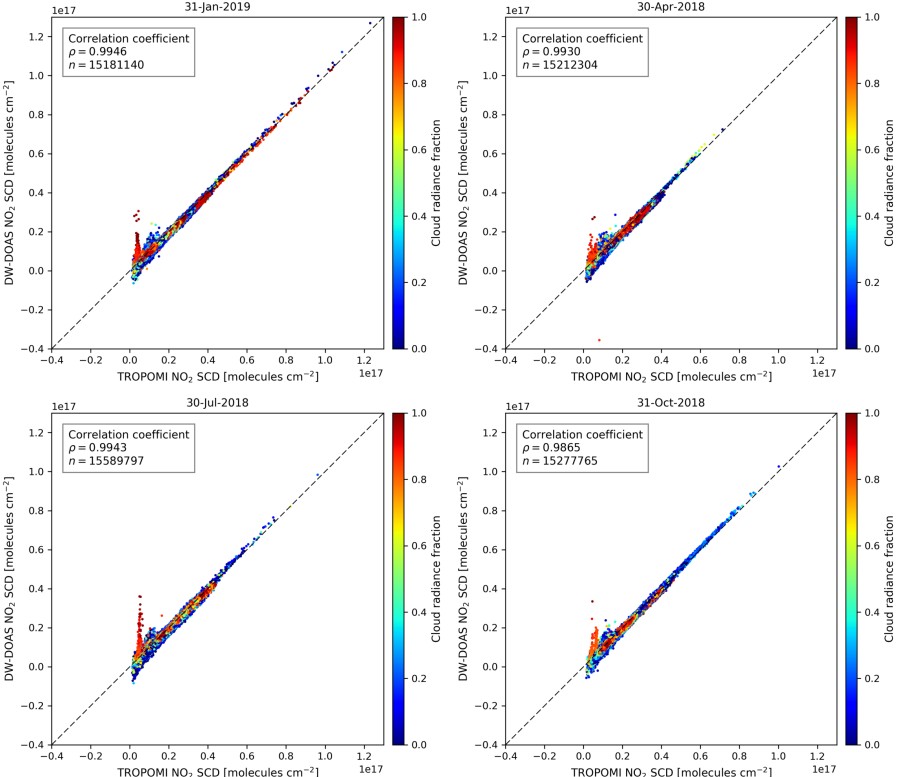

**Figure 10.** Correlation plots of DW-DOAS and TROPOMI global NO$_2$ SCD data used in Figure 9.

## 3.4 Method limitations

Some of the challenges of using DW-DOAS for NO$_2$ come from using limited spectral information to retrieve a relatively weak absorber. One limitation is the increased sensitivity to random noise, as seen in the retrieval results and demonstrated by the SCD statistical uncertainty estimations. Another limitation is the higher sensitivity to interfering species, since there is not

5 enough spectral information to completely separate out the gas of interest from the other species. However, the effect of this can be minimised by optimising the wavelength selection.

Furthermore, DW-DOAS has particular limitations that stem from the use of discrete wavelengths in combination with the DOAS retrieval technique. Firstly, one of the basic premises of DOAS is the removal of broadband structures from the reflectance spectra using a polynomial, typically 4th or 5th order. As explained in section 2.3.1, using a polynomial of such a

10 high degree would cause the retrieval to underestimate the NO$_2$ SCD, so this is limited to a 2nd-order polynomial. However, sometimes this is not enough to remove complex surface albedo or scattering broadband structures. These residual structures might be the underlying cause behind some of the higher SCD differences between DW-DOAS and the OMI and TROPOMI reference products, and can be minimised by selecting channels that are close together so that they can be approximated by a 2nd-order polynomial.



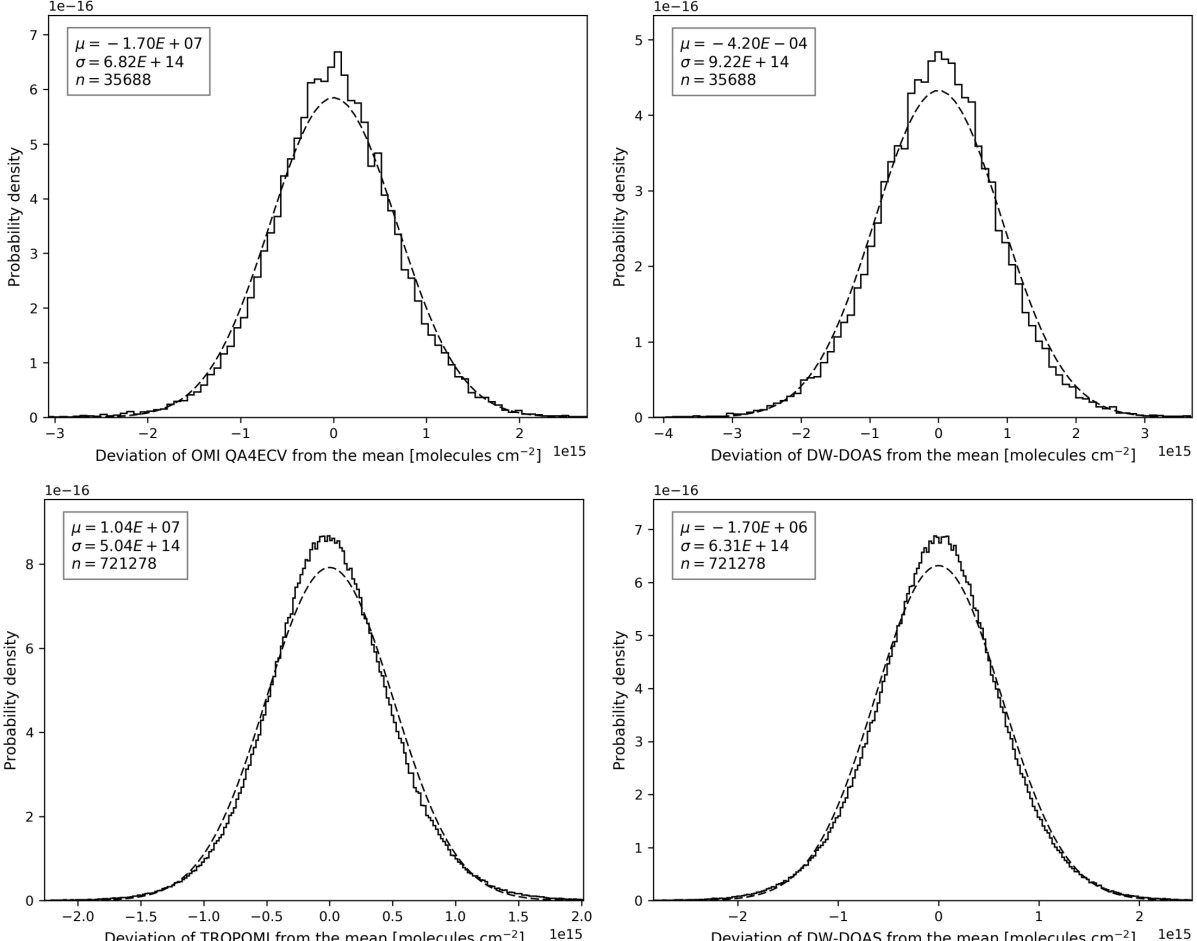

**Figure 11.** Histogram of the deviation of the NO$_2$ SCDs from the box mean for OMI (top) and TROPOMI (bottom), for DW-DOAS (right) and the corresponding reference products (left). OMI data has not been screened for clouds; TROPOMI data has been screened using the quality assurance value (qa > 0.5) from the level 2 NO$_2$ product. The data shown are from January 2005 (OMI) and January 2019 (TROPOMI).

Finally, wavelength calibration using a high-resolution solar reference is an important step in DOAS retrievals because even a small wavelength shift can cause retrieval errors. With discrete-wavelength data it is not possible to use the Fraunhofer lines of the solar spectrum for the wavelength calibration. While this is a shortcoming, it is anticipated that small wavelength shifts do not have as big an impact as they are for hyperspectral DOAS precisely because the spectral channels are sparse and not
5   contiguous, and because the filters are wider.





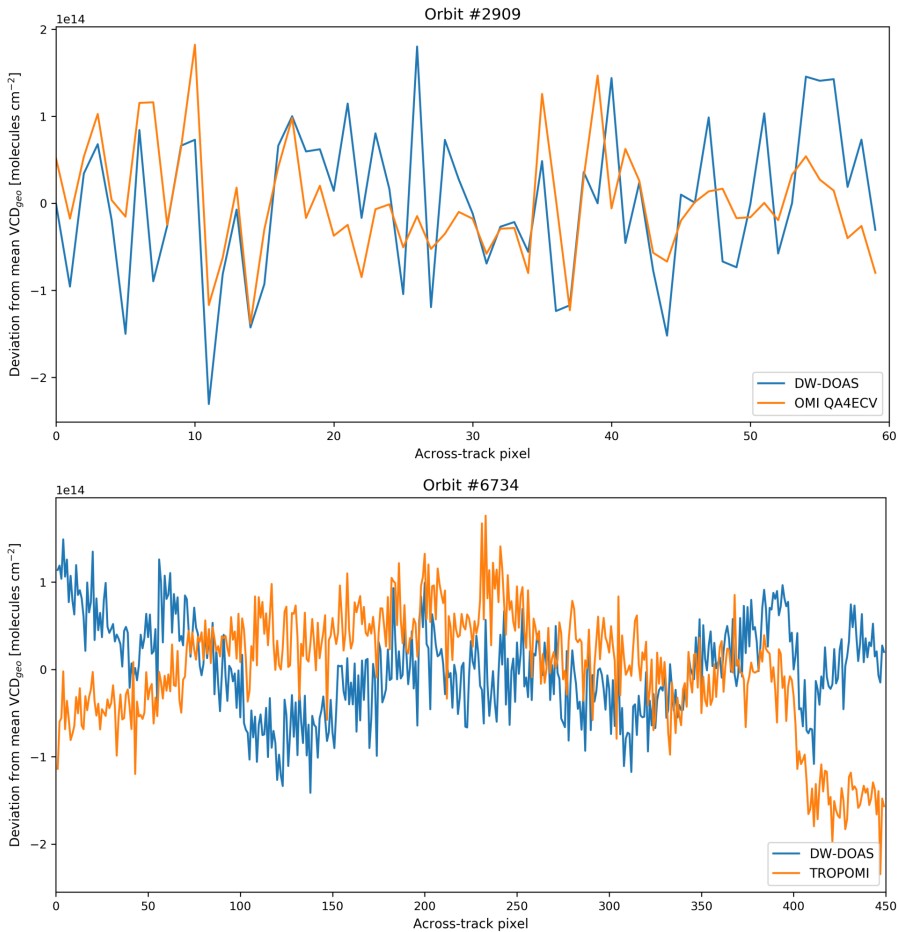

**Figure 12.** Striping sensitivity of DW-DOAS and the corresponding reference product for (top) OMI and (bottom) TROPOMI data for one orbit over the Pacific Ocean in the latitude range 30° S – 5° N.

### 3.5 Considerations for future instruments

The DW-DOAS results we have presented are promising and the method has potential to be applied to new satellite instrument designs. However, several aspects need to be considered before it can be implemented in an operational instrument:

- **Reference spectrum.** The DW-DOAS method has only been tested using a solar spectrum as the reference ($I_0$) for the DOAS fit. However, Earthshine radiance spectra could be used instead, as demonstrated by Anand et al. (2015). Using these spectra would simplify the instrument design by removing the need for a solar diffuser and a solar measurement mode; it would cancel out some instrumental effects and reduce the effect of the Ring structures in the retrieval. Moreover, in theory a synthetic solar spectrum could also be used. Nevertheless, all options come with drawbacks, so a sensitivity study would be needed to find the approach that provides the best results.





- **Wavelength calibration and filter response function.** As discussed in Section 3.4, it is not possible to perform a wavelength calibration for DW-DOAS using a solar reference spectrum owing to the lack of spectral information. Thus, a mechanism for in-flight monitoring of the spectral response of the filters would be critical, since these need to be known accurately to convolve the absorption cross sections.

- **Cloud retrieval.** To use DW-DOAS operationally a cloud retrieval would be needed to identify cloudy pixels. In other retrieval algorithms using visible spectra this is performed using knowledge of the $O_2$-$O_2$ slant column, which can be derived from its absorption cross section peak at ~477 nm (Veefkind et al., 2016). However, it is not possible to retrieve $O_2$-$O_2$ with the wavelengths proposed in this work because they are optimised for $NO_2$. Therefore, further work is needed to find a suitable solution, for example, by adding a few channels to detect the aforementioned $O_2$-$O_2$ peak.

- **AMF calculations.** This work has evaluated the performance of the $NO_2$ SCD retrieval. However, that is only the first of the three steps in an $NO_2$ tropospheric vertical column retrieval, which is the final product for the typical end user. The other two steps are the stratospheric-tropospheric $NO_2$ separation (e.g. model assimilation, Boersma et al. (2011)), and the conversion of SCDs into vertical column densities (VCDs) using air mass factors (AMFs; Palmer et al., 2001). These two steps are mostly independent from the SCD fit, so in principle no major differences are expected for DW-DOAS. However, further work is needed to test these and ensure that any retrieval-dependent sensitivities are understood before DW-DOAS is implemented in an operational instrument. This is particularly true for very high spatial resolutions, where the surface albedo might be a problem for DW-DOAS owing to the polynomial limitation.

## 4 Conclusions and further work

We have developed a method, DW-DOAS, to perform $NO_2$ slant column density retrievals using only 10 discrete spectral channels and the DOAS technique. It has been tested using OMI and TROPOMI datasets and found to produce results that are comparable to the reference level 2 products, with a mean difference of ~5 % for OMI QA4ECV and ~11 % for TROPOMI. However, DW-DOAS has higher uncertainties, which are due to a higher sensitivity to noise, and it is more sensitive to striping. While there is a high correlation ($r > 0.99$) of the DW-DOAS results with the reference level 2 products, some spatial variabilities are found. The largest differences are seen over water, near the equator, in the Sahara desert, and in clear-sky areas with low $NO_2$ SCDs. In addition, the centre of the swath presents higher differences. The cause of these is unknown but low $NO_2$ concentrations and short light paths might play a role.

The main advantage of the DW-DOAS method over existing DOAS retrievals is the need for comparatively little spectral information, which makes the retrieval faster and would allow potential instruments designs with high spatial resolution. Limitations of the method include higher sensitivity to broadband structures such as surface albedo; higher sensitivity to noise, which means a higher SNR is required; inability to perform a wavelength calibration using a high-resolution solar reference; and the ability to retrieve only $NO_2$, although with further work it might be possible to retrieve other species by adding a small number of channels.


Despite the shortcomings, our results show that the DW-DOAS method has potential. It could be used in future satellite instruments to allow simpler designs and low-cost constellations for air quality monitoring at high resolution. This type of constellation could be a good complement to existing high-budget hyperspectral instruments such as OMI and TROPOMI, for example, for the detection of small scale $NO_2$ hotspots, which could be identified from space and investigated further using

*in situ* instruments. Furthermore, DW-DOAS could potentially be used for faster retrievals (e.g. for near-real time processing) for hyperspectral data from existing instruments. Processing speed is especially important for higher data volumes expected by future high-resolution instruments.

Next steps for this work shall include optimising the DW-DOAS method, particularly the channel selection, including the selection of optimal centre wavelengths, number of channels, filter widths, and a comprehensive sensitivity study. Moreover,

the practicalities of implementing this method on a real instrument need further assessment: wavelength calibration, reference spectrum ($I_0$) for the DOAS retrieval (Earthshine/solar spectrum), cloud retrieval, and the next stages of the $NO_2$ retrieval (tropospheric/stratospheric separation, and AMF calculation).

## Appendix A: Absolute differences between DW-DOAS and OMI and TROPOMI

Figure A1 shows the absolute differences between the geometric column densities (i.e. SCDs scaled with the geometric AMFs)

as retrieved by DW-DOAS and the OMI/TROPOMI L2 reference products.

*Competing interests.* The authors declare that they have no conflict of interest.

*Acknowledgements.* This research was funded by the CENTA Doctoral Training Programme (Grant NE/L002493/1) as part of the Natural Environmental Research Council (UK), in partnership with Thales Alenia Space UK. This work follows the outcomes of the project "High-resolution Anthropogenic Pollution Imager (HAPI) on an OmniSat Platform", funded by CEOI-ST under the UKSA's NSTP Flagship

programme (Grant RP10G0348C01).

The OMI Level 1B data used in this work (OML1BRVG.003) are publicly available at https://disc.gsfc.nasa.gov/datasets?page=1&source=AURA%20OMI. The OMI QA4ECV $NO_2$ Level 2 data were obtained from http://temis.nl/airpollution/no2col/no2regioomi_qa.php. The TROPOMI Level 1B data (L1B_RA_BD4 and L1B_IR_UVN) and some of the $NO_2$ Level 2 data (offline products from July and October 2018, and January 2019) were obtained from https://s5phub.copernicus.eu/dhus/#/home. The TROPOMI $NO_2$ offline reprocessed Level 2

data from April 2018 were obtained from http://temis.nl/airpollution/no2col/no2regio_tropomi.php. The maps shown in Figures 3, 5, 7, and 9 were plotted using the cartopy Python package (Met Office, 2010 - 2015) with default coast lines, which uses freely available Natural Earth map data. This research used the ALICE High Performance Computing Facility at the University of Leicester.

We thank Jos van Geffen for kindly providing the auxiliary files (absorption cross sections) used in the OMI and TROPOMI reference products, and we are grateful to him and Folkert Boersma for advising on the analysis of the results (Tables 2 and 3).



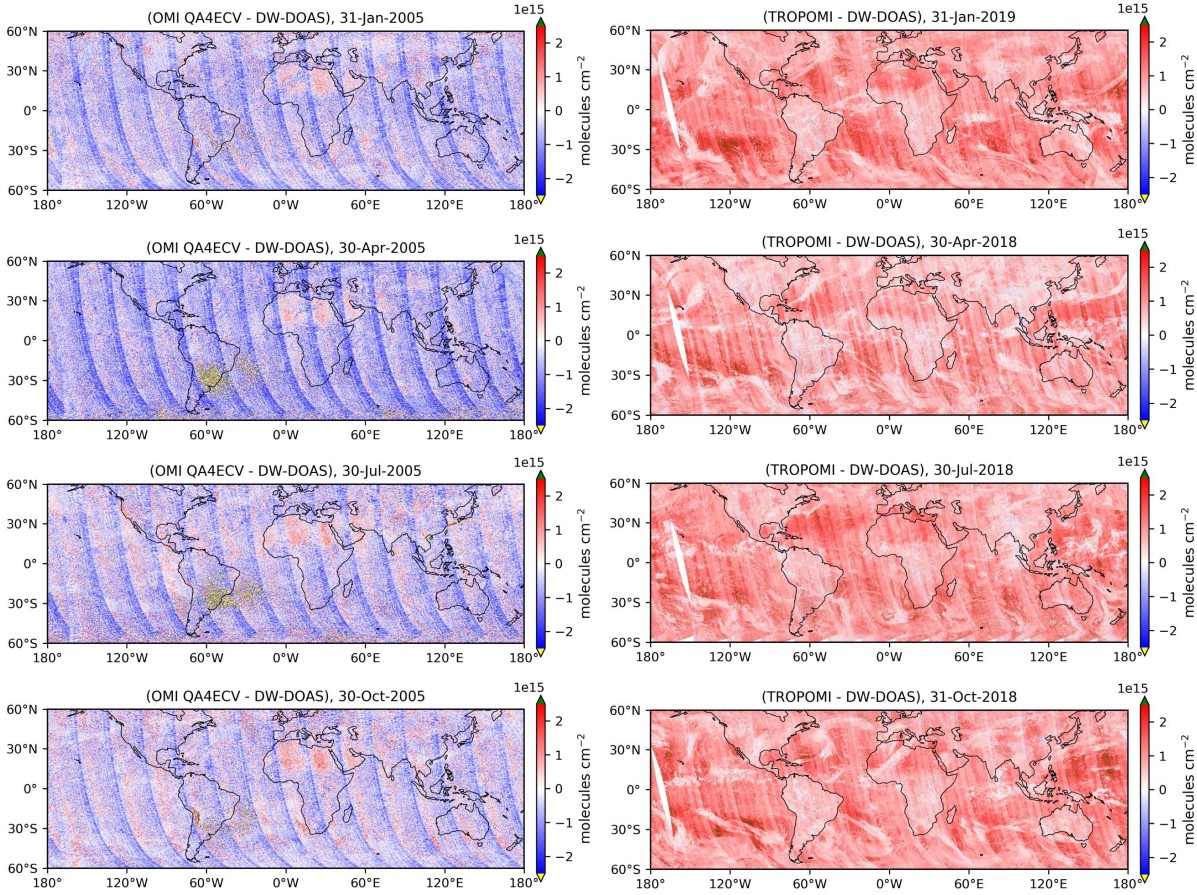

**Figure A1.** Absolute differences between DW-DOAS and: (left) OMI QA4ECV, and (right) TROPOMI, for the data shown in Figures 5 and 9. The differences are expressed in terms of the SCD scaled with the geometric AMF. The latitudes are limited to [60° S, 80° N].

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
