# Peer review of "Discrete-wavelength DOAS $\text{NO}_2$ slant column retrievals from OMI and TROPOMI"

_Atmospheric Measurement Techniques, 2019_

## Referee Comment (RC1) · Anonymous Referee #1 · 20 Sep 2019

The manuscript "Discrete-wavelength DOAS NO2 slant column retrievals from OMI and TROPOMI" describes a NO2 retrieval algorithm based on the DOAS method with discrete spectral channels. The idea of discrete channels has been applied for ozone retrieval, and its potential for NO2 retrieval is shown in this manuscript, addressing the advantage of simpler instrumental design. The retrieval is implemented for OMI and TROPOMI data with good agreement with respect to reference products (5% difference for OMI and 11% difference for TROPOMI). Critical issues like the selection of discrete channels, uncertainties, and limitations are discussed. The topic of the manuscript is within the scope of AMT.

My major concern with this manuscript is the verification or validation. In principal the overall quality of a retrieval needs to be evaluated by comparisons with independent

satellite retrievals or by comparisons with correlative ground-based measurements (e.g., direct sun measurements from Pandora). Since the authors have shown only specific days as examples for comparisons with reference datasets, the retrieval quality can hardly be analyzed without a longer timeseries reprocess of OMI and TROPOMI slant column data and additional comparisons, which are particularly important for discrete-wavelength DOAS (with no wavelength calibration). Therefore I recommend that the authors include more verification or validation results to check for possible systematic bias or temporal drift of differences.

Another general request is that please follow the standard use of mathematics notation in the literature. For instance, an upright bold symbol needs to be used in the equation and text to make it clear where vectors and matrices are discussed, and also a matrix is usually written enclosed in square brackets.

The absolute differences are plotted in the appendix, but the analysis in the manuscript only focuses on the relative differences. For instance, "the largest differences around the equator" is actually only valid for the relative difference figure 5 (due to the small absolute values). Please add more discussions of the absolute differences.

Specific comments

P2 L21 Generally the observation is separated into in situ measurements and remote sensing measurements, and the remote sensing technique can be further separated into space-based and ground-based category.

P4 L3 What has been decreased by 0.5%? Do you mean 0.5% of the degradation?

P4 L19 Please give the full name of SNR.

P7 L5 x shall be a column vector.

P8 Table 1. Should the fitting window for DW-DOAS be 425-450 nm (425-450 nm appears also in Table 2)?

[Figure]

P9 L26 Why are the negative biases related to the differences in the fitting window? Theoretically the differences in the fitting window shall affect only the scatter of the NO2 columns (i.e. noise) but not the fitted value of NO2 column.

P10 L17 What is the reason of more outliers for lower cloud radiance fractions for OMI and the opposite for TROPOMI? Also what is the impact of cloud height on these plots? Generally the retrieved column should depend strongly on the bulk height of clouds. High clouds mask the signal from surface NO2 while for low clouds the satellite observations remain sensitive to the NO2 in the free troposphere.

P12 L14 The spatial patterns might be related to the intensity offset correction. The intensity offset correction included in the TROPOMI reference algorithm compensates spectral structures of liquid water, vibrational Raman scattering on H2O molecules, and possible instrumental issue, leading to a difference over the cloud-free tropical ocean. Please refer to the QA4ECV report for more discussion. In addition, the pattern can also be seen a bit from the OMI absolute difference plot, but it is overwhelmed in the relative difference plot. Therefore more analysis about the absolute results has been required (see the major comments).

---

## Referee Comment (RC2) · Anonymous Referee #2 · 29 Oct 2019

General comments

This manuscript presents a simplified NO2 slant column retrieval approach, which makes use of a limited number of wavelengths in an otherwise classical DOAS retrieval framework. The approach is tested on sample data from the OMI and TROPOMI sensors, and results are discussed in terms of their consistency with standard retrievals. It is concluded that retrievals based on strongly sub-sampled spectra (only 10 wavelengths are used) still provide good NO2 slant columns. Although this result is not surprising as such (given the high quality of the original measurements), the small reduction of the noise on the retrieved slant columns is in my view a bit unexpected and worth pointing out (and possibly explain). So we come with the conclusion that retrieving NO2 columns from 10 spectral points is feasible. There are however a number

of drawbacks and limitations in doing so, and one may wonder whether the potential advantages of reducing the spectral information would actually compensate these drawbacks.

The fundamental motivation behind the study relies on the postulate that reducing the spectral information would allow to simplify instrumental design (of future satellite missions) leading to potentially improved spatial resolution at low cost. Statements along these lines are given at several places in the manuscript, but without any further elaboration, e.g. what kind of instrumental solution could be adopted? More importantly, key requirements on spectral accuracy and stability that would need to be considered for such a design are not mentioned at all. It is basically assumed that spectral performances equivalent to those of OMI and TROPOMI can "easily" be obtained with low-cost imaging systems suitable for integration on small satellites. To my opinion, the lack of such a discussion significantly limits the relevance and impact of the study. I therefore recommend publication only if these questions are better addressed in a major revision of the manuscript.

Specific comments

Pg. 2, line 25: not all sources of tropospheric NO2 are anthropogenic in nature. Please complete.

Pg. 2, line 43: in addition to in-situ and satellite techniques, also ground-based remote sensing constitute a key component of the atmospheric composition monitoring system. This includes e.g. the Network for the Detection of Atmospheric Composition Change (NDACC) or the emerging Pandonia/PGN network.

Pg. 2, line 48: current satellite instruments are limited in resolution, but TROPOMI is already doing much better than OMI. This should already be mentioned here, with a mention that ultimate resolutions in the range of 1x1 km2 are needed to allow for individual source identification.

Pg. 2, line 56: The current resolution of TROPOMI at true nadir is 3.5 x 5.5 km2.

Pg. 3, line 71: the Brewer instrument is cited here as an example for a NO2 measuring system based on a few wavelengths; however it is well-known that Brewer NO2 measurements are dramatically lacking sensitivity. This was actually at the origin of the development of the Pandora instrument, which uses simple (low-cost) grating spectrometers to (strongly) improve the quality of NO2 column measurements.

Pg. 3, line 75: what are the "specific viewing geometries" that prevent usage of the NO2 camera for space applications? Please clarify.

Pg. 4, line 116: describe in short the interpolation method used by Buscela, and its added value for this study

Pg. 5, line 123: this introductory paragraph is a bit misleading. To my understanding the critical aspect of selecting appropriate spectral channels for NO2 fitting is not related to the complexity of the radiative transport, but only to the nature of the differential cross-sections and the presence of interfering species.

Pg. 5, line 131: replace "mean optical depth" by "differential optical depth" (or difference in optical depth)

Pg. 6, Figure 2: how important is it to include liquid water cross-sections in the fitting? In the spectral range of interest, this cross-sections seem to be very unstructured and may correlate strongly with the polynomial function.

Pg. 6, line 144 (very minor comment): the choice of "discrete wavelength DOAS" as a name could in fact be questioned, since fundamentally all DOAS schemes use discrete wavelengths (it is just that in your case, their number is smaller)

Pg. 9, line 199: how can local variations in surface albedo explain differences between retrievals from same satellite pixels? Please clarify the meaning of this statement.

Pg. 13, Figure 5: why such a discontinuity in the NO2 map of 30 Oct 2005 (at 20°S)?

This looks like an artefact apparent in both QA4ECV and DW-DOAS results (considering the difference plot)

Pg. 13, line 265: the explanation given for the striping problem is inexact. Stripes in NO2 SCDs are not related to inhomogeneous illumination of the entrance slit, but merely due to small issues in spectral calibration, dark current correction and detector sensitivity (see e.g. Boersma et al., 2018)

Pg. 13, line 260-264: the small difference in noise despite the large difference in the number of spectral points considered for the retrieval is somehow a (good) surprise to me. I would have expected much larger differences. It would be interesting to further investigate and explain the fundamental reason for this lack of dependence of the noise on the number of spectral points.

Pg. 14, Figure 6: my guess is that most of the outliers in the correlation plots are related to the SAA. This could be easily verified by excluding the SAA area from the comparison.

Pg. 17, line 285: this sentence is maybe not necessary here, since it is a repetition of what has been said before.

Pg. 18, line 290: as already mentioned in my general comments, this part of the manuscript lacks more details on the instrumental challenges (or difficulties) associated to the potential new instruments. In particular, for DOAS retrievals of tropospheric NO2, it would be essential that the instrument allow for perfect spatial co-location of the 10 wavelengths and that all of them are recorded simultaneously. Any deviation with respect to these requirements might lead to spectral distortion affecting the accuracy of the slant column retrieval.

Pg. 19, line 299: as pointed out in the paper, possible wavelength calibration inaccuracies are a potential source of error. It would be interesting to investigate the sensitivity of the algorithm to such errors. This would also provide an idea of the associated

requirement on instrument design.

Pg. 20, line 306: note that O4 retrieval is not needed for cloud fraction retrieval, but only for cloud top height retrieval. For a sensor working at high spatial resolution, a good working option would be to rely only on cloud free pixels (without the need for a cloud correction).

————————————————

---

## Author Comment (AC1) · 17 Jan 2020

**RESPONSES TO REVIEWERS**

Ms. Ref. No.: Atmos. Meas. Tech. Discuss., doi:10.5194/amt-2019-252.

Title: Discrete-wavelength DOAS $NO_2$ slant column retrievals from OMI and TROPOMI

**Response to Anonymous Referee #1**

*The manuscript "Discrete-wavelength DOAS $NO_2$ slant column retrievals from OMI and TROPOMI" describes a $NO_2$ retrieval algorithm based on the DOAS method with discrete spectral channels. The idea of discrete channels has been applied for ozone retrieval, and its potential for $NO_2$ retrieval is shown in this manuscript, addressing the advantage of simpler instrumental design. The retrieval is implemented for OMI and TROPOMI data with good agreement with respect to reference products (5% difference for OMI and 11% difference for TROPOMI). Critical issues like the selection of discrete channels, uncertainties, and limitations are discussed. The topic of the manuscript is within the scope of AMT.*

*My major concern with this manuscript is the verification or validation. In principal the overall quality of a retrieval needs to be evaluated by comparisons with independent satellite retrievals or by comparisons with correlative ground-based measurements (e.g., direct sun measurements from Pandora). Since the authors have shown only specific days as examples for comparisons with reference datasets, the retrieval quality can hardly be analysed without a longer time series reprocess of OMI and TROPOMI slant column data and additional comparisons, which are particularly important for discrete-wavelength DOAS (with no wavelength calibration). Therefore I recommend that the authors include more verification or validation results to check for possible systematic bias or temporal drift of differences.*

Thank you for your comment. We agree that further validation would be needed if the aim was to establish a new method to retrieve $NO_2$. However, the work presented in this paper is intended only as a proof of concept rather than a comprehensive validation of a new product. Thus, for this purpose the reference OMI and TROPOMI level 2 products are considered as the "truth" and our retrieval results validated against them. We selected four days from different seasons to get a range of solar angles and prevailing atmospheric conditions, and we used global data to factor in a wide range of atmospheric scenarios and spatial differences. The differences between our retrieval and the reference products are consistent across all the data with the exception of some spatial differences, which we have already discussed in the text. It is expected that the main factor affecting a time series would be noise from the degradation of the instrument, which would manifest in the form of higher scatter in the DW-DOAS retrieval. Therefore, the authors feel that further validation is beyond the scope of this paper, but are currently working on more comprehensive sensitivity analyses and validation which will be the focus of the next paper. We have clarified in the text that the focus of the paper is only to perform a proof of concept.

*Another general request is that please follow the standard use of mathematics notation in the literature. For instance, an upright bold symbol needs to be used in the equation and text to make it clear where vectors and matrices are discussed, and also a matrix is usually written enclosed in square brackets.*

We have now corrected the mathematical expressions.

*The absolute differences are plotted in the appendix, but the analysis in the manuscript only focuses on the relative differences. For instance, "the largest differences around the equator" is actually only valid for the relative difference figure 5 (due to the small absolute values). Please add more discussions of the absolute differences.*

Thanks for spotting this. We have added more discussion of the absolute differences.

**Specific comments**
*P2 L21 Generally the observation is separated into in situ measurements and remote sensing measurements, and the remote sensing technique can be further separated into space-based and ground-based category.*

Very good point. We have added the ground-based remote sensing technique and included a couple of examples provided by Anonymous Referee #2.

*P4 L3 What has been decreased by 0.5%? Do you mean 0.5% of the degradation?*

This refers to the performance of OMI's visible channel, which has had a radiometric degradation of ~0.5%. We have now modified the statement to make it clearer.

*P4 L19 Please give the full name of SNR.*

We added the full name and put the acronym in brackets, i.e. signal-to-noise ratio (SNR).

*P7 L5 x shall be a column vector.*

Corrected.

*P8 Table 1. Should the fitting window for DW-DOAS be 425-450 nm (425-450 nm appears also in Table 2)?*

Thanks for spotting this inconsistency. Yes, even though the first wavelength used in DW-DOAS sits around 430 nm, the fitting window should read '425-450 nm'. This range was selected from the literature. The concept of "fitting window" does not apply in the same way as it does in hyperspectral

DOAS retrievals, since we are not using continuous spectra. It should rather be interpreted as a spectral range from where we select our ten discrete wavelengths. We have modified Table 1 and added a line discussing the different interpretation of the fitting window in the context of DW-DOAS.

*P9 L26 Why are the negative biases related to the differences in the fitting window? Theoretically the differences in the fitting window shall affect only the scatter of the $NO_2$ columns (i.e. noise) but not the fitted value of $NO_2$ column.*

Wider fitting windows have traditionally been used to achieve higher signal-to-noise ratios. However, when such windows are used there is a higher chance of introducing other spectral signatures that are not accounted for in the retrieval, resulting in systematic biases (Richter et al., 2011). Evidence of this effect specific to the two windows used in this work can be found in Figure 11 of van Geffen et al. (2015), where changing the fitting window from 405 – 465 nm to 425 – 450 nm causes the fitted SCD to change by up to +0.5E15 molecules $cm^{-2}$. We have added a line explaining this.

*P10 L17 What is the reason of more outliers for lower cloud radiance fractions for OMI and the opposite for TROPOMI? Also what is the impact of cloud height on these plots? Generally the retrieved column should depend strongly on the bulk height of clouds. High clouds mask the signal from surface $NO_2$ while for low clouds the satellite observations remain sensitive to the $NO_2$ in the free troposphere.*

We agree with this assessment, and have amended the manuscript accordingly. However, we do not believe that a fair comparison can be made between the two results. For instance, it must be noted that the OMI and TROPOMI observations are over a decade apart and so will be subject to very different cloud structures. Additionally, TROPOMI has a smaller pixel size and so will experience very different cloud radiance fractions to OMI (see Krijger et al, 2007). Finally, there may also be inherent differences between the cloud top heights observed by both instruments based on the different retrieval algorithms they employ; OMI retrieves this parameter using the $O_2$-$O_2$ absorption feature at 477 nm (Veefkind et al, 2016), while TROPOMI makes use of the $O_2$-A band in its operational retrieval (Loyola et al, 2018). In addition, the QA4ECV product for OMI includes an intensity offset correction, which is not included in the TROPOMI product, and that may explain some of the differences over the ocean (Oldeman, 2018).
In addition, we have updated Figure 8 as there was a plotting error whereby the x and y axes were swapped.

*P12 L14 The spatial patterns might be related to the intensity offset correction. The intensity offset correction included in the TROPOMI reference algorithm compensates spectral structures of liquid water, vibrational Raman scattering on $H_2O$ molecules, and possible instrumental issue, leading to a difference over the cloud-free tropical ocean. Please refer to the QA4ECV report for more discussion. In addition, the pattern can also be seen a bit from the OMI absolute difference plot, but it is*

*overwhelmed in the relative difference plot. Therefore more analysis about the absolute results has been required (see the major comments).*

Thanks for the suggestion. The TROPOMI product doesn't include an intensity offset correction. However, the OMI QA4ECV product does and we agree that it could well explain some of the spatial differences. We have now added more discussion about this and the absolute results.

**References:**

Krijger, J. M., van Weele, M., Aben, I., and Frey, R.: Technical Note: The effect of sensor resolution on the number of cloud-free observations from space, Atmos. Chem. Phys., 7, 2881–2891, https://doi.org/10.5194/acp-7-2881-2007, 2007.

Loyola, D. G., Gimeno García, S., Lutz, R., Argyrouli, A., Romahn, F., Spurr, R. J. D., Pedergnana, M., Doicu, A., Molina García, V., and Schüssler, O.: The operational cloud retrieval algorithms from TROPOMI on board Sentinel-5 Precursor, Atmos. Meas. Tech., 11, 409–427, https://doi.org/10.5194/amt-11-409-2018, 2018.

Oldeman, A.: Effect of including an intensity offset in the DOAS NO2 retrieval of TROPOMI, Internship report, R1944-SE, Eindhoven University of Technology/KNMI, Eindhoven, May 2018, https://kfolkertboersma.files.wordpress.com/2018/06/report_oldeman_22052018.pdf, (last access: 10 January 2020), 2018.

Richter, A., Begoin, M., Hilboll, A., and Burrows, J.P.: An improved $NO_2$ retrieval for the GOME-2 satellite instrument, Atmospheric Measurement Techniques, 4, 1147–1159, https://doi.org/10.5194/amt-4-1147-2011, 2011.

van Geffen, J. H. G. M., Boersma, K. F., Van Roozendael, M., Hendrick, F., Mahieu, E., De Smedt, I., Sneep, M., and Veefkind, J. P.: Improved spectral fitting of nitrogen dioxide from OMI in the 405–465 nm window, Atmos. Meas. Tech., 8, 1685–1699, https://doi.org/10.5194/amt-8-1685-2015, 2015.

Veefkind, J. P., de Haan, J. F., Sneep, M., and Levelt, P. F.: Improvements to the OMI $O_2$–$O_2$ operational cloud algorithm and comparisons with ground-based radar–lidar observations, Atmos. Meas. Tech., 9, 6035–6049, https://doi.org/10.5194/amt-9-6035-2016, 2016.

---

## Author Comment (AC2)

**RESPONSES TO REVIEWERS**

Ms. Ref. No.: Atmos. Meas. Tech. Discuss., doi:10.5194/amt-2019-252. Title: Discrete-wavelength DOAS NO2 slant column retrievals from OMI and TROPOMI

**Response to Anonymous Referee #2**

This manuscript presents a simplified NO2 slant column retrieval approach, which makes use of a limited number of wavelengths in an otherwise classical DOAS retrieval framework. The approach is tested on sample data from the OMI and TROPOMI sensors, and results are discussed in terms of their consistency with standard retrievals. It is concluded that retrievals based on strongly sub-sampled spectra (only 10 wavelengths are used) still provide good NO2 slant columns. Although this result is not surprising as such (given the high quality of the original measurements), the small reduction of the noise on the retrieved slant columns is in my view a bit unexpected and worth pointing out (and possibly explain). So we come with the conclusion that retrieving NO2 columns from 10 spectral points is feasible.

There are however a number of drawbacks and limitations in doing so, and one may wonder whether the potential advantages of reducing the spectral information would actually compensate these drawbacks. The fundamental motivation behind the study relies on the postulate that reducing the spectral information would allow to simplify instrumental design (of future satellite missions) leading to potentially improved spatial resolution at low cost. Statements along these lines are given at several places in the manuscript, but without any further elaboration, e.g. what kind of instrumental solution could be adopted? More importantly, key requirements on spectral accuracy and stability that would need to be considered for such a design are not mentioned at all. It is basically assumed that spectral performances equivalent to those of OMI and TROPOMI can "easily" be obtained with low-cost imaging systems suitable for integration on small satellites. To my opinion, the lack of such a discussion significantly limits the relevance and impact of the study. I therefore recommend publication only if these questions are better addressed in a major revision of the manuscript.

Thank you for your comment. Our intention was to demonstrate that the concept of retrieving NO2 using only 10 discrete wavelengths is feasible and that accuracy comparable to existing level 2 NO2 products can theoretically be achieved. We absolutely agree that in practice the concept is more complex and there are implementation challenges that must be overcome, e.g. co-registration of the spectral bands. While the study was deliberately discussed in generic terms, i.e. independent of any specific instrument solution, we acknowledge that the manuscript would benefit from more discussion about implementation challenges and potential instrument solutions. We have now added such discussion in paragraph 3 of the conclusions. Regarding the key requirements on spectral stability and accuracy, those are the focus of the next study, which we are currently working on to derive such requirements from sensitivity analyses.

**Specific comments**

**Pg. 2, line 25: not all sources of tropospheric NO2 are anthropogenic in nature. Please complete.**

We did not intend to suggest that all sources of tropospheric NO2 are anthropogenic, but rather that anthropogenic emissions are the main contributor. We have clarified this in the text and completed the statement with examples of natural sources of tropospheric NO2.

Pg. 2, line 43: in addition to in-situ and satellite techniques, also ground-based remote sensing constitute a key component of the atmospheric composition monitoring system. This includes e.g. the Network for the Detection of Atmospheric Composition Change (NDACC) or the emerging Pandonia/PGN network.

Thanks for pointing this out, also highlighted by Anonymous Referee #1. We have now mentioned the ground-based remote sensing technique and given the suggested examples.

*Pg. 2, line 48: current satellite instruments are limited in resolution, but TROPOMI is already doing much better than OMI. This should already be mentioned here, with a mention that ultimate resolutions in the range of 1x1 km2 are needed to allow for individual source identification.*

Thanks for your comment. We now give the example of TROPOMI instead of OMI, and mention the spatial resolution requirement for point source identification.

Pg. 2, line 56: The current resolution of TROPOMI at true nadir is 3.5 x 5.5 km2.

We have now clarified that the stated resolution is at true nadir.

Pg. 3, line 71: the Brewer instrument is cited here as an example for a  $NO_2$  measuring system based on a few wavelengths; however it is well-known that Brewer  $NO_2$  measurements are dramatically lacking sensitivity. This was actually at the origin of the development of the Pandora instrument, which uses simple (low-cost) grating spectrometers to (strongly) improve the quality of  $NO_2$  column measurements.

Thanks for pointing this out. We have added the lack of sensitivity as another drawback of the Brewer spectrometer.

*Pg.* 3, line 75: what are the "specific viewing geometries" that prevent usage of the  $NO_2$  camera for space applications? Please clarify.

The algorithm used in the AOTF-based NO2 camera as described in Dekemper et al. (2016) relies on clear-sky pixels being present in the scene for background subtraction. In addition, the sequential sampling of wavelengths poses a limitation to the speed at which they can be registered, making the

retrieval challenging for non-static scenes. These drawbacks make the NO2 camera unsuitable for nadir-viewing space applications. We have clarified this in the text.

*Pg. 4, line 116: describe in short the interpolation method used by Bucsela, and its added value for this study*

The method used by Bucsela et al. (2006) calculates the interpolated spectrum using the high-resolution solar reference spectrum as follows:

$$F(\lambda + d\lambda) = \frac{F(\lambda)}{[F_0(\lambda + d\lambda)/F_0(\lambda)]}$$

Where *F* is the measured spectrum,  $F_0$  is the solar reference spectrum,  $\lambda$  is the original wavelength grid of *F*, and  $\lambda + d\lambda$  is the new wavelength grid. In Bucsela et al. (2006) the irradiance spectrum is interpolated onto the radiance wavelength grid, whereas in our work we interpolate the radiance onto the irradiance wavelength grid to match what is done for the OMI and TROPOMI L2 products.

This method is an improvement over other approaches (e.g. linear or spline) as it reduces interpolation errors related to the sampling rate. However, this improvement is not expected to be significant for instruments like OMI and TROPOMI where undersampling is not a problem. We have updated the text with this clarification.

Pg. 5, line 123: this introductory paragraph is a bit misleading. To my understanding the critical aspect of selecting appropriate spectral channels for  $NO_2$  fitting is not related to the complexity of the radiative transport, but only to the nature of the differential cross-sections and the presence of interfering species.

We agree with your assessment in the case of traditional DOAS NO2 fits. However, discrete-wavelength DOAS is more sensitive to scattering and albedo effects than traditional DOAS because the polynomial models the broadband component of the reflectance less accurately. This is why the fitting interval must be narrow enough to minimise the effect of the broadband component. The cross sections and the interfering species are still key aspects but in the case of discrete-wavelength DOAS the complexity of the radiative transport also plays an important role. Nonetheless, we acknowledge that the paragraph is misleading and have updated it.

Pg. 5, line 131: replace "mean optical depth" by "differential optical depth" (or difference in optical depth)

Corrected to "differential optical depth".

Pg. 6, Figure 2: how important is it to include liquid water cross-sections in the fitting? In the spectral range of interest, this cross-sections seem to be very unstructured and may correlate strongly with the polynomial function.

We did some tests and concluded that including the liquid water cross section does not make much difference. However, we included it in the fit to match the reference retrieval settings as closely as possible.

*Pg.* 6, line 144 (very minor comment): the choice of "discrete wavelength DOAS" as a name could in fact be questioned, since fundamentally all DOAS schemes use discrete wavelengths (it is just that in your case, their number is smaller)

We agree, good point. We considered different names and concluded that "discrete-wavelength DOAS" was the one that best described the retrieval approach while still being clear and short. Nonetheless, we welcome suggestions for alternative names that might be more suitable.

**Pg. 9, line 199: how can local variations in surface albedo explain differences between retrievals from same satellite pixels? Please clarify the meaning of this statement.**

As we discuss in a previous comment, discrete-wavelength DOAS is more sensitive to albedo effects than traditional DOAS. Therefore, we think that one possible explanation for the bigger retrieval differences in the smaller pixels might be related to the different sub-pixel variability of the albedo due to the size of the pixel. In other words, we would generally expect less albedo variability within smaller pixels and this might mean that some stronger spectral features might be present compared to the bigger pixels. These strong spectral features would result in higher retrieval differences between DW-DOAS and the reference level 2 products. We have clarified this in the text.

**Pg. 13, Figure 5: why such a discontinuity in the NO2 map of 30 Oct 2005 (at 20°S)?**

This discontinuity is also present in the reference QA4ECV NO2 Level 2 product (see http://temis.nl/airpollution/no2col/no2regioomi\_qa.php?Region=9&Year=2005&Month=10&Day=30). We don't exactly know what causes it but it is not present in the tropospheric NO2 column map, so it is likely a combination of stratospheric NO2 and processes/elements involved in the air mass factor (AMF) calculation such as atmospheric scattering.

**References:**

Bucsela, E., Celarier, E., Wenig, M., Gleason, J., Veefkind, J., Boersma, K., and Brinksma, E.: Algorithm for NO2 vertical column retrieval from the ozone monitoring instrument, IEEE Transactions on Geoscience and Remote Sensing, 44, 1245, https://doi.org/10.1109/TGRS.2005.863715, 2006.

Dekemper, E., Vanhamel, J., Opstal, B. V., and Fussen, D.: The AOTF-based NO2 camera, Atmospheric Measurement Techniques, 9, 6025–6034, https://doi.org/10.5194/amt-9-6025-2016, 2016.

---

## Author Response (AR2)

**RESPONSES TO REVIEWERS (05/03/2020)**

Ms. Ref. No.: Atmos. Meas. Tech. Discuss., doi:10.5194/amt-2019-252.

Title: Discrete-wavelength DOAS $NO_2$ slant column retrievals from OMI and TROPOMI

*Reviewers' comments in blue*. Responses in black.

**Response to Associate Editor**

*1) In the discussion of the interpolation of the radiance spectra, two things should be noted:*

 *a) F_0 is the high resolution irradiance, convoluted with the instrument slit function*

 *b) Applying this scheme to radiances instead of irradiances is problematic as the shape of the Fraunhofer lines changes as result of rotational Raman scattering in the atmosphere.*

a) In this case F_0 is actually a high-resolution solar reference spectrum, rather than the irradiance, as per the interpolation method described in Bucsela et al. (2006). We have clarified what each term means in the text.

b) Initially, we interpolated the irradiance to the radiance for OMI and TROPOMI, but we changed it later for OMI to match what is done for the QA4ECV $NO_2$ retrieval (Zara et al., 2018). We have now stated this in the text.

*2) I agree with the reviewer that the large scatter in your global correlation plots is probably due to the SAA. It would be worthwhile to filter the data before creating the scatter plots as these fits are not expected to be good.*

Good point. We have now created new correlation plots for Figures 6 and 10 filtering out the SAA pixels in the red box shown in the map below (Figure 1). It does reduce the scatter significantly.

[Figure]

**Figure 1**. Example map from the manuscript (Fig. A1) showing the area affected by the SAA. The red box shows the pixels that have been filtered out for the correlation plots (latitudes [6º S, 40º S], longitudes [70º W, 16º W]).

*3) Please make sure that figures 4, 6 and 8 use the same logic with respect to choice of what to plot as x, and what as y-axis.*

Thanks for spotting this. We have changed Figure 4 to match the others.

*4) In several places, the discussion of vibrational Raman scattering contains the formulation "over water". While I assume that you refer to the fact that measurements over water are affected by this signal, it still could be misunderstood as vibrational Raman scattering in the air over water instead on water molecules within the ocean.*

We have clarified this in the text.

**Response to Anonymous Referee #2**

*1) p.2, l.5: add fires as an important natural source of $NO_2$.*

Added.

*2) p.2. l.35: remove the last sentence, and add "(ultimately in the range of 1x1 km2) " between "... spatiotemporal resolution" and "is required...".*

Corrected.

*3) Figure 6: is the noise at low column values not related to the SAA? -> maybe this region could be filtered out?*

Good point. Yes, some of it is related to the SAA. As per your suggestion and the Associate Editor's comment, we have now filtered out the pixels in that area. We have updated all correlation plots in Figures 6 and 10. See response to comment 2) of the Associate Editor.

*4) p.16, l.8: I think that the TROPOMI $NO_2$ algorithm uses a $NO_2$-specific cloud algorithm (so not the operational cloud algorithm reference in Loyola et al., 2018). You may refer to the TROPOMI NO2 ATBD or equivalent reference if available.*

Thanks for pointing this out. According to the TROPOMI $NO_2$ ATBD, they use the cloud pressure $p_c$ from the FRESCO-S product and calculate the cloud fraction and cloud radiance fraction from the $NO_2$ window at 440 nm. It is also mentioned that apart from FRESCO-S, a new cloud algorithm is under development at DLR.

We have replaced the Loyola et al. (2018) reference with the TROPOMI $NO_2$ ATBD, as the cloud algorithm using in the $NO_2$ retrieval is explained there.

*5) p.23, l.28: add "potentially" between "could" and "be".*

Added.

**References:**

[revised manuscript text omitted]